# Stress, anxiety, depression and sleep disturbance among healthcare professional during the COVID-19 pandemic: An umbrella review of 72 meta-analyses

Mohammed Al Maqbali[1], Ahmad Alsayed[2], Ciara Hughes[3], Eileen Hacker[4], Geoffrey L. Dickens [5,6]*

1 Fatima College of Health Sciences, Al Ain, United Arab Emirates, 2 Faculty of Pharmacy, Department of Clinical Pharmacy and Therapeutics, Applied Science Private University, Amman, Jorden, 3 Institute of Nursing and Health Research School of Health Sciences, Ulster University, Belfast, United Kingdom, 4 University of Texas MD Anderson Cancer Center, Houston, Texas, United States of America, 5 Midwifery and Health Faculty of Health and Life Sciences, Mental Health Nursing Department of Nursing, Northumbria University, Newcastle-Upon-Tyne, United Kingdom, 6 Adjunct Professor Western Sydney University, Parramatta, NSW, Australia

* geoffrey.dickens@northumbria.ac.uk

**Data Availability Statement:** All data are in the paper and/or supporting information files.

## Abstract

The outbreak of SARS-CoV-2, which causes COVID-19, has significantly impacted the psychological and physical health of a wide range of individuals, including healthcare professionals (HCPs). This umbrella review aims provide a quantitative summary of meta-analyses that have investigated the prevalence of stress, anxiety, depression, and sleep disturbance among HCPs during the COVID-19 pandemic. An umbrella review of systematic reviews and meta-analyses reviews was conducted. The search was performed using the EMBASE, PubMed, CINAHL, MEDLINE, PsycINFO, and Google Scholar databases from 01st January 2020 to 15th January 2024. A random-effects model was then used to estimate prevalence with a 95% confidence interval. Subgroup analysis and sensitivity analyses were then conducted to explore the heterogeneity of the sample. Seventy-two meta-analyses involved 2,308 primary studies were included after a full-text review. The umbrella review revealed that the pooled prevalence of stress, anxiety, depression, and sleep disturbance among HCPs during the COVID-19 pandemic was 37% (95% CI 32.87–41.22), 31.8% (95% CI 29.2–34.61) 29.4% (95% CI 27.13–31.84) 36.9% (95% CI 33.78–40.05) respectively. In subgroup analyses the prevalence of anxiety and depression was higher among nurses than among physicians. Evidence from this umbrella review suggested that a significant proportion of HCPs experienced stress, anxiety, depression, and sleep disturbance during the COVID-19 pandemic. This information will support authorities when implementing specific interventions that address mental health problems among HCPs during future pandemics or any other health crises. Such interventions may include the provision of mental health support services, such as counseling and peer support programs, as well as the implementation of organizational strategies to reduce workplace stressors.

**Funding:** The author(s) received no specific funding for this work.

**Competing interests:** The authors have declared that no competing interests exist.

## 1. Introduction

In December 2019, the coronavirus disease 2019 (COVID-19) pandemic emerged in Wuhan, China. The disease quickly spread worldwide, and the WHO declared a global health emergency in March 2020 [1]. Due to the COVID-19 pandemic, many countries implemented various measures to prevent the spread of the disease. These included implementing a partial or complete lockdown and social distancing strategies of varying intensity. The measures taken by these countries also affected the livelihood of individuals, an occurrence which might directly or indirectly also increase psychological morbidities. Undoubtedly, pandemics have a long history of impacting physical and mental health for different population groups, and HCPs are typically the most affected group in terms of bearing the burden of these illnesses [2]. In addition, several researchers have shown that work-related psychological disorders, including stress, anxiety, depression, and burnout, had already negatively affected the healthcare system before the COVID-19 pandemic, leading to low-quality care and high malpractice litigation [3–5].

As a result of the pandemic, HCPs experienced various changes in their personal and professional lives. For some, these included being given more responsibility, having to re-learn how to effectively control the infection, and dealing with the emotional impact of caring for infected and dying COVID-19 patients [6]. The alteration in their work environment, as well as the likelihood that they might acquire the infection themselves, can also affect their personal mental health. It is almost inevitable that the experiences of HCPs went through during the pandemic put them at heightened risk of stress, anxiety, depression, and sleep disturbance [7,8]. It is important to understand the effects of the pandemic on the mental health and well-being of HCPs in order to help plan strategies to prevent these individuals from experiencing detrimental effects, and to ensure that they can continue to deliver healthcare services.

During previous viral outbreaks including the Severe Acute Respiratory Syndrome (SARS) and the Middle East Respiratory Syndrome (MERS) epidemics, HCPs were placed under extraordinary amounts of pressure [9,10]. Indeed, evidence suggested that HCPs suffered from high levels of stress, anxiety, depression and sleep disturbance during these outbreaks [11,12]. A high prevalence of mental health problems can adversely impact the quality of life of HCPs, increase disability, turnover, absenteeism, and errors, and can deleteriously affect patient outcomes which may lead to low patient satisfaction [13]. Further, it might increase suicidal ideation or self-harming among HCPs [14].

In the present review, four phenomena were addressed. Sleep disturbance refers to a range of sleep-related problems, including disruptions in the body's natural sleep-wake cycle, insufficient or poor-quality sleep, and sleep disorders [15]. The anxiety symptoms were defined as a state of excessive fear that translates to behavioural disturbances [16]. Major depressive disorder is a set of symptoms that includes depressed mood, loss of pleasure or interest, fatigue, changes in sleep and activity levels, and other symptoms, with a minimum duration of two weeks and at least five or more symptoms present according to the Diagnostic and Statistical Manual of Mental Disorders (DSM-5) [16]. Cohen et al., [17] define stress as "the degree to which individuals appraise situations in their lives as stressful". In this umbrella review, stress, anxiety, depression, and sleep disturbance symptoms were defined based on the validated scales/questionnaires that assess each phenomenon in the original studies.

Several primary studies and, subsequently, systematic reviews and meta-analyses have been carried out to identify the prevalence of mental health problems among HCPs during the COVID-19 pandemic. Additionally, three umbrella reviews of meta-analyses [18–20] have been published previously, but the number of meta-analyses included in both cases did not exceed twenty. Since their publication, further meta-analyses have estimated the prevalence of

stress, anxiety, depression, and sleep disturbance during the COVID-19 pandemic. The advantages of umbrella reviews include their ability to provide a comprehensive analysis of the literature, in this case about the prevalence of various mental disorders in HCPs during the COVID-19 pandemic. In addition, the results can then be used to make policy-level decisions to improve the quality of clinical care in terms of making clinical risk predictions and can inform future research priorities. Therefore, the aim of this umbrella review is to quantify meta-analytic findings aimed at estimating the prevalence symptoms of stress, anxiety, depression, and sleep disturbance among HCPs during the COVID-19 pandemic.

## 2. Methods

The umbrella review and meta-analysis were carried out according to the Preferred Reporting Items for Systematic Reviews and Meta-analyses (PRISMA) guidelines [21]. The review protocol was registered with the International Prospective Register of Systematic Reviews (PROSPERO) database and can be accessed online (CRD42022364721).

### 2.1 Search strategy

A systematic search was conducted to identify relevant meta-analyses in various electronic databases published between 1$^{st}$ January 2020 and 15$^{th}$ January 2024. The databases searched were PubMed, CINAHL, MEDLINE, EMBASE, PsycINFO, and Google Scholar. The search terms strategy used Medical Subject Headings (MeSH) and free text words with Boolean operators and truncations (AND/OR/NOT). The key search terms included (MH "Coronavirus Infections+") OR "COVID-19" OR "COVID" OR "coronavir* OR "Coronavirus" OR "SARS-COV2" AND "Health care provider" OR "health care professional" OR "healthcare provider*" OR (MH "Nurses+") OR (MH "Medical Staff") OR (MH "Physician") OR (MH "Medical Doctor") OR (MH "Staff Nurses") OR "nursing staff" OR "health personnel or health professional or nurse" OR "health personnel or health professional or nurse" AND "Stress" OR "post-traumatic stress disorder" OR "panic disorder" OR "obsessive compulsive disorder OR "anxi*" OR (MH "Anxiety Disorders+") OR (MH "Anxiety+") OR (MH "Depression+") OR "depress*" OR (MH "Affective Symptoms+") OR (MH "Affective Disorders+") OR (MH "Bipolar Disorder+") OR "affective" OR "mood" OR "mental" OR (MH "Mental Disorders+") OR (MH "Mental OR "psycho*" OR (MH "Insomnia+") OR (MH "Circadian Rhythm+") OR (MH "Sleep Disorders+") OR (MH "Insomnia+") OR (MH "Sleep +") AND "Systematic Review" OR "Meta-Analysis" OR "Meta-Analytic". Additionally, the reference lists were searched to find any other studies.

### 2.2 Study selection

Two reviewers (A.M.; A.A.) independently extracted the data from the search, scrutinizing all titles and abstracts for eligibility against the inclusion and exclusion criteria. A third reviewer (G.D.) was available to resolve any disagreements through discussion. Systematic reviews incorporating meta-analyses were included according to the following criteria. The studies: (1) examined the prevalence of stress or anxiety or depression or sleep disturbance symptoms; (2) presented results for HCPs as a group or separately (e.g., nurses or physicians only); further, studies involving non-HCPs must have presented results for HCPs separately and not pooled with non-HCPs.; (3) were conducted during the COVID-19 pandemic; (4) were published in English; (5) involved a systematic review with meta-analysis. Studies were excluded if (1) these consisted of a systematic review without meta-analysis; (2) consisted of a literature review or a narrative review (3) the participants were general population or non-HCPs.

## 2.3 Quality assessment

The methodological quality assessment of each meta-analysis was blindly rated by two reviewers using the Assessment of Multiple Systematic Reviews (AMSTAR-2) tool [22]. This scale consists of 16 items that evaluate the risk of bias of a systematic review. Items 1, 3, 5, 6, 10, 13, 14, and 16 are evaluated with either a "Yes" or "No" response. Items 2, 4, 7, 8, and 9 are evaluated with "Yes," "Partial Yes," or "No" responses. Items 11, 12, and 15 are evaluated with "Yes," "No," or "No meta-analysis conducted" responses. The overall rating can be rated as "High," "Moderate," "Low," or "Critically low."

## 2.4 Credibility of evidence

The credibility of the evidence of each association provided was evaluated by the Fusar-Poli and Radua [23] classification criteria. The level of evidence as convincing (class I) when specific criteria were met, including more than 1000 cases, $p<10^{-6}$, $I^2$ higher than 50%, 95% prediction intervals excluding the null, no small-study effects, and no publication bias. If the number of cases exceeded 1000, $p<10^{-6}$, the largest study showed a statistically significant effect, but not all class I criteria were satisfied; the evidence level was considered highly suggestive (class II). When there were over 1000 cases, $p<10^{-3}$, but no other class I or II criteria were met, the evidence level was termed suggestive (class III). If no class I-III criteria were met, the evidence level was classified as weak (class IV). The fourth level, termed weak evidence (class IV), included associations with a $p \leq 0.05$, but these associations did not meet the criteria for class I, class II, or class III. The fifth level, denoted as non-significant (NS), comprised associations with a $p > 0.05$.

## 2.5 Data analyses

There are two methods exist for deriving effect size estimates from existing meta-analyses. The first approach involves conducting a meta-analysis on the effect size estimates taken from individual studies included in multiple prior meta-analyses [24]. However, this method demands significant time and resources. Furthermore, it contradicts the primary purpose of an umbrella review because it requires return to the original studies.

The second approach employs a statistical technique to efficiently summarize data from previous meta-analyses without the need to go back to the individual studies. This method relies solely on the summary effect sizes and their associated variances provided in the original meta-analyses [25]. It calculates an overall effect size for the combined meta-analyses by computing a weighted average of the summary effect sizes, with the weights determined by the inverse of the variances [26]. This approach is similar to the methods used in meta-analyses of primary studies. Although the second approach (combining summary effect sizes) may not achieve the same level of precision as the first method (combining all individual studies), empirical tests have confirmed its ability to generate a statistically valid estimate for the overall effect size [27,28]. In this umbrella review, we employed the second approach, which entailed the utilization of aggregate data derived from the meta-analyses.

The analyses were conducted using R software, version 4.3.1 (R Foundation for Statistical Computing), with packages used 'meta' [29], 'metafore' [30] and 'metaumbrella'[31]. Pooled estimates prevalence with 95% Confidence Intervals (CIs) was conducted using random effect models, and the results were reported on a forest plot. In addition, the I-squared ($I^2$) test was used to assess the statistical heterogeneity of the included meta-analyses. A value of $I^2 < 25\%$ was considered low, 25–50% moderate, and $> 50\%$ high [32]. Subgroup analyses were performed when there were at least four meta-analyses per subgroup.

Publication bias was assessed using Egger's test with a p < 0.10 indicates a statistically significant small-study effect [33]. Statistical significance was set at p<0.05. If publication bias was identified, trim and fill methods were used to adjust the publication bias [34]. A sensitivity analysis was conducted in which individual meta-analyses were systematically removed one at a time to assess how they affected the overall combined prevalence of the remaining meta-analyses [35], with the aim of clarifying the stability and reliability of the finding [36].

## 3. Result

A total of 1,987 papers were identified through the database search. Out of these, 1,843 were excluded at the abstract and title screening stage for the following reasons: 786 were duplicates, 443 did not include a meta-analysis, 392 lacked information about prevalence, 139 lacked information about HCP status, and 83 were not conducted during the period of the COVID-19 pandemic. A further 72 papers were excluded during the full text review process. As a result, 72 meta-analyses were eligible for umbrella review (Fig 1).

### 3.1 General characteristics of the studies included

The included 72 meta-analyses [37–108]. Fifty-four of the meta-analyses dealt with HCPs in general, whereas two meta-analyses reported the situation only with regard to physicians or nurses [44,59], three meta-analyses dealt with nurses [49,79,91], and one dealt with physicians [88], while 17 meta-analyses included a mixed population (General and HCPs),(Only data specifically related to healthcare professionals were included in the umbrella review analysis). The most commonly used statistical software was STATA (n = 32), R (n = 17) and comprehensive meta-analysis (n = 11). Twenty-four meta-analyses used the Newcastle–Ottawa scale to assess the quality of the studies. Forty-six meta-analyses included mixed studies from different countries, twenty-five meta-analyses were conducted in specific geographical areas: 10 for China, five for India, four for Asia two for Bangladesh and Ethiopia and one for each of the following: Egypt, South Asia, and Vietnam. The detailed characteristics of the studies including meta-analyses are shown in Table 1.

### 3.2 Quality appraisal

Each meta-analysis was assessed using the AMSTAR-2 tool. Twenty- nine meta-analyses were classified as moderate quality and thirty-seven as low quality. Only six meta-analyses were classified as critically low quality [37,44,45,76,97,101].

### 3.3 Prevalence of stress

Stress was reported in 42 meta-analyses among HCPs. The estimation of prevalence for stress varied between 11% [72] and 66.6% [58] (Fig 2: Forest Plots). The pooled prevalence of stress from was 37% (292,245/854,852 participants, 95% CI 32.87–41.22) with 95% PI: 14.86–66.3. There was significant heterogeneity between meta-analyses when it came to estimating the prevalence of stress ($p < 0.0001$, $I^2 = 99.9\%$). In the subgroup analysis, the prevalence of anxiety among nurses was determined to be 42.6% (n = 5; 95% CI = 30.49–55.27, $I^2 = 99\%$), as shown in Fig 3: Forest Plots. However, the analysis for physicians was not conducted due to an insufficient number of available meta-analyses. In sensitivity analysis, none of the meta-analyses resulted in changes to the pooled prevalence estimates greater than a 2%. The prevalence rate estimates for stress were considered to be suggestive evidence (class III) (Seen Table 2).

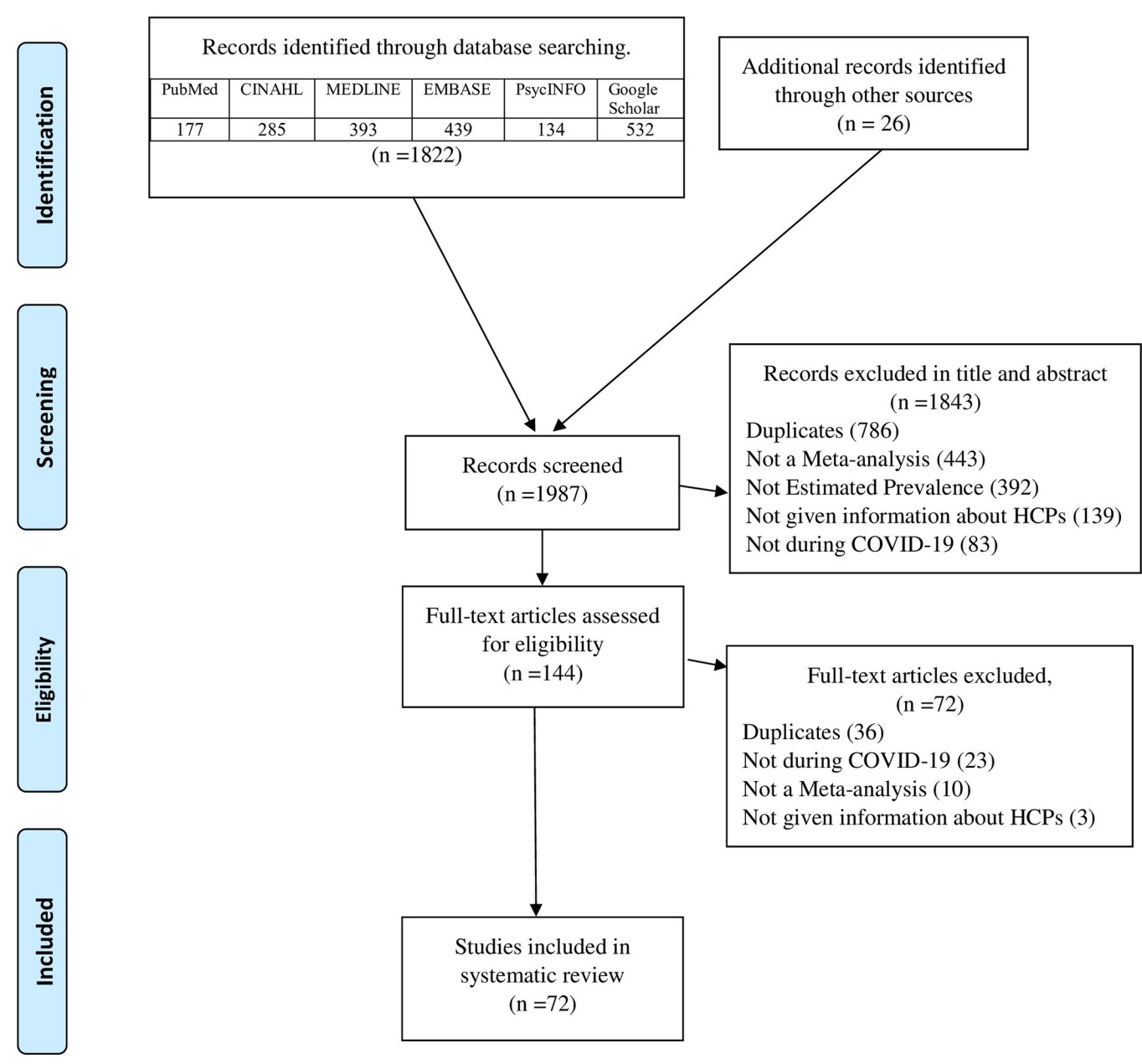

**Fig 1. PRISMA diagram.**

### 3.4 Prevalence of anxiety

Fifty-five meta-analyses estimated the prevalence of anxiety among HCPs, and ranged from 11.4% [40] to 71.9% [58]. The pooled prevalence of anxiety was 31.8% (734,036/2,310,774 participants, 95% CI 29.2–34.61) with 95% PI: 15.24–54.83 (Fig 4: Forest Plots) among all HCPs and there was considerable heterogeneity ($p< 0.0001$, $I^2 = 99.9\%$). In the subgroup analyses, in terms of professional status, the pooled prevalence of anxiety was 31.6% (n = 12; 95% CI = 28.33–35.14, $I^2 = 99\%$) and 26.3% (n = 9; 95% CI = 22.89–30.10, $I^2 = 99\%$) for nurses, and physicians respectively (Figs 5 and 6: Forest Plots). A sensitivity analysis, specifically a leave-

**Table 1. Characteristics of the included meta-analyses (N = 72).**

| Study | Total Study Included | Outcome | Population | Study Included | Prevalence | 95% CI | I² | Events | Sample size | Analysis Software | Model | Appraisal Tool | AMSTAR-2 Quality | Remarks |
|---|---|---|---|---|---|---|---|---|---|---|---|---|---|---|
| 1 (Batra et al. 2020) [38] | 65 | Stress | All | 17 | 40.3 | 31.4–50.0 | 99.1 | 6543 | 16235 | CMA | REM | NIH | Moderate | HCPs |
| | | Anxiety | All | 46 | 34.4 | 29.5–39.7 | 99.1 | 17749 | 51596 | | | | | |
| | | | Nurses | 8 | 39.3 | 27.5–52.5 | 98.9 | 8606 | 21899 | | | | | |
| | | | Physicians | 8 | 32.5 | 21.9–45.2 | 99.0 | 7117 | 21899 | | | | | |
| | | Depression | All | 46 | 31.8 | 26.8–37.2 | 99.2 | 16906 | 53164 | | | | | |
| | | | Nurses | 9 | 42.4 | 30.4–5.4 | 99.0 | 9301 | 21936 | | | | | |
| | | | Physicians | 9 | 39.1 | 27.3–52.2 | 98.4 | 8577 | 21936 | | | | | |
| 2 (Krishnamoorthy et al. 2020) [39] | 23 | Stress | All | 5 | 33 | 19–50 | 98.4 | 1986 | 6017 | STATA | REM | NOS | Moderate | Mixed |
| | | Anxiety | All | 16 | 24 | 16–32 | 99.6 | 9201 | 38337 | | | | | |
| | | Depression | All | 16 | 25 | 19–32 | 99.4 | 9425 | 37701 | | | | | |
| | | Sleep Disturbance | All | 5 | 43 | 28–59 | 99.2 | 2010 | 4675 | | | | | |
| 3 (Pappa et al. 2020) [41] | 13 | Anxiety | All | 12 | 23.2 | 17.8–29.1 | 99 | 7367 | 31756 | MetaXL | REM | NOS | Moderate | HCPs |
| | | | Nurses | 6 | 25.8 | 19.2–33.0 | 98 | 5160 | 19999 | | | | | |
| | | | Physicians | 6 | 21.7 | 15.3–29 | 97 | 4340 | 19999 | | | | | |
| | | Depression | All | 10 | 22.8 | 15.1–31.5 | 99.6 | 7071 | 31014 | | | | | |
| | | | Nurses | 5 | 30.3 | 18.2–43.8 | 99.5 | 5679 | 18742 | | | | | |
| | | | Physicians | 5 | 25.4 | 16.6–35.2 | 98 | 4760 | 18742 | | | | | |
| | | Sleep Disturbance | All | 5 | 34.3 | 27.5–41.5 | 98 | 2935 | 8558 | | | | | |
| 4 (Qiu et al. 2020) [42] | 52 | Sleep Disturbance | All | 52 | 39.2 | 36.0–427 | 98.3 | 12446 | 31749 | R | REM | Loney | Low | HCPs/China |
| | | | Nurses | 41 | 39.4 | 35.7–43.5 | 98.4 | 9878 | 25070 | | | | | |
| | | | Physicians | 3 | 34.2 | 12.6–55.8 | 99.2 | 720 | 2104 | | | | | |
| 5 (Salari et al. 2020a) [44] | 7 | Sleep Disturbance | Nurses | 6 | 34.8 | 24.8–46.4 | 97.4 | 2526 | 7259 | CMA | REM | STROBE | Critically Low | HCPs |

(Continued)

Table 1. (Continued)

| Study | Total Study Included | Outcome | Population | Study Included | Prevalence | 95% CI | I² | Events | Sample size | Analysis Software | Model | Appraisal Tool | AMSTAR-2 Quality | Remarks |
|---|---|---|---|---|---|---|---|---|---|---|---|---|---|---|
| 6 (Salari et al. 2020b) [45] | 29 | | Physicians | 5 | 41.6 | 27.7–57 | 97.3 | 2286 | 5496 | | | STROBE | Critically Low | HCPs/Frontline |
| | | Stress | All | 8 | 36.4 | 18.3–59.5 | 99 | 1293 | 3551 | CMA | REM | | | |
| | | Anxiety | All | 17 | 27 | 20.1–35.3 | 95.5 | 2987 | 11062 | | | | | |
| | | Depression | All | 15 | 20.6 | 13.1–30.9 | 99.1 | 2196 | 10658 | | | | | |
| 7 (Allan et al. 2020) [37] | 9 | Stress | All | 9 | 23.4 | 16.3–31.2 | 96 | 970 | 4147 | R | REM | NIH | Critically Low | HCPs |
| 8 (Salazar de Pablo et al. 2020) [46] | 15 | Stress | All | 3 | 29.9 | 8.9–65.2 | 99.7 | 2030 | 6789 | CMA | REM | MMAT | Moderate | Mixed |
| | | Anxiety | All | 4 | 22.2 | 12.7–35.8 | 99 | 1713 | 7716 | | | | | |
| | | Depression | All | 4 | 17.9 | 6.7–40.1 | 99.6 | 1381 | 7716 | | | | | |
| | | Sleep Disturbance | All | 3 | 44.5 | 38.2–50.9 | 93 | 1553 | 3490 | | | | | |
| 9 (Ren et al. 2020) [43] | 5 | Anxiety | All | 4 | 27 | 12–43 | 99.4 | 1308 | 4843 | STATA | REM | AHRQ | Low | Mixed |
| | | Depression | All | 3 | 25 | 4–45 | 99.7 | 1166 | 4663 | | | | | |
| 10 (Pan et al. 2020) [40] | 7 | Anxiety | All | 7 | 11.4 | 7.1–15.8 | 99 | 848 | 7441 | STATA | REM | STROBE | Low | Mixed |
| 11 (Zhang et al. 2021) [82] | 26 | Stress | All | 8 | 42.1 | 26.0–60.0 | 99.5 | 3866 | 9183 | R | REM | NIH | Moderate | HCPs/China |
| | | Anxiety | All | 23 | 27.0 | 21.0–34.0 | 99.0 | 5791 | 21447 | | | | | |
| | | Depression | All | 18 | 26.2 | 21.0–33.0 | 98.6 | 4930 | 18818 | | | | | |
| | | Sleep Disturbance | All | 8 | 35.5 | 28.0–42.0 | 98.3 | 4174 | 11758 | | | | | |
| 12 (Adibi et al. 2021) [48] | 19 | Anxiety | All | 19 | 30.5 | 25.6–35.4 | 98.4 | 6669 | 21866 | STATA | REM | NG | Low | HCPs |
| 13 (Al Maqbali et al. 2021) [49] | 93 | Stress | Nurses | 40 | 43 | 37–49 | 98 | 11625 | 27034 | CMA | REM | NOS | Moderate | Nurses |
| | | Anxiety | Nurses | 73 | 37 | 32–41 | 99 | 30178 | 81561 | | | | | |
| | | Depression | Nurses | 62 | 35 | 31–39 | 99 | 26947 | 76992 | | | | | |
| | | Sleep Disturbance | Nurses | 18 | 43 | 36–50 | 96 | 4600 | 10697 | | | | | |
| 14 (Alimoradi et al. 2021) [50] | 62 | Sleep Disturbance | All | 62 | 43 | 39–47 | 99.3 | 25521 | 59350 | STATA | REM | NOS | Low | Mixed |
| 15 (Bareeqa et al. 2021) [51] | 8 | Depression | All | 8 | 31.5 | 20.7–43.5 | 99.3 | 3234 | 10267 | OMA | REM | NOS | Low | Mixed/China |

(Continued)

**Table 1.** (Continued)

| Study | Total Study Included | Outcome | Population | Study Included | Prevalence | 95% CI | I² | Events | Sample size | Analysis Software | Model | Appraisal Tool | AMSTAR-2 Quality | Remarks |
|---|---|---|---|---|---|---|---|---|---|---|---|---|---|---|
| 16 (Cénat et al. 2021) [52] | 27 | Stress | All | 4 | 21.0 | 9.0–43.0 | 99.7 | 1444 | 6878 | R | REM | JBI | Moderate | Mixed |
| | | Anxiety | All | 23 | 15.9 | 12.2–20.3 | 99 | 5895 | 37076 | | | | | |
| | | Depression | All | 18 | 14 | 11.0–17.0 | 95.3 | 4781 | 34151 | | | | | |
| | | Sleep Disturbance | All | 6 | 16 | 8.0–30.0 | 99.8 | 993 | 6209 | | | | | |
| 17 (Ching et al. 2021) [53] | 148 | Stress | All | 40 | 36.4 | 23.2 | 49.7 | 12380 | 34010 | OMA | REM | STROBE | Low | HCPs/Asia |
| | | Anxiety | All | 117 | 39.7 | 34.3–45.1 | 99.8 | 39557 | 99639 | | | | | |
| | | Depression | All | 98 | 37.5 | 33.8–41.3 | 99.5 | 38861 | 103628 | | | | | |
| 18 (Dutta et al. 2021) [57] | 33 | Stress | All | 17 | 37.7 | 24.0–52.3 | 100 | 10269 | 27238 | MetaXL | REM | NOS | Low | HCPs |
| | | Anxiety | All | 31 | 32.5 | 26.4–39 | 99 | 7628 | 23472 | | | | | |
| | | Depression | All | 30 | 32.4 | 25.6–39.3 | 99 | 12200 | 37655 | | | | | |
| | | Sleep Disturbance | All | 11 | 36.6 | 36.6–48.3 | 99 | 3810 | 10411 | | | | | |
| 19 (Hao et al. 2021) [60] | 20 | Stress | All | 5 | 25.6 | 11.8–39.4 | 99 | 790 | 3085 | R | REM | AHRQ | Low | HCPs |
| | | Anxiety | All | 16 | 28.6 | 22.4–36.4 | 99 | 2864 | 10015 | | | | | |
| | | | Nurses | 7 | 23.5 | 15.8–35 | 99 | 1437 | 6113 | | | | | |
| | | Depression | All | 14 | 24.1 | 16.2–32.1 | 99 | 1915 | 7948 | | | | | |
| | | | Nurses | 5 | 25 | 12.7–37.4 | 96 | 423 | 1692 | | | | | |
| | | Sleep Disturbance | All | 5 | 44.1 | 31.3–57 | 98 | 1607 | 3643 | | | | | |
| 20 (Yan et al. 2021) [81] | 35 | Stress | All | 4 | 56 | 32–79 | 97 | 428 | 765 | R | REM | STROBE | Low | HCPs/China |
| | | Anxiety | All | 29 | 41 | 35–47 | 98 | 6439 | 15704 | | | | | |
| | | Depression | All | 20 | 27 | 20–32 | 94 | 5680 | 21038 | | | | | |
| | | Sleep Disturbance | All | 8 | 41 | 33–50 | 98 | 3065 | 7476 | | | | | |
| 21 (Liu et al. 2021) [65] | 33 | Stress | All | 5 | 30.6 | 9.1–65.9 | 98 | 979 | 3200 | CMA | REM | NOS | Low | Mixed |

(Continued)

**Table 1.** (Continued)

| Study | Total Study Included | Outcome | Population | Study Included | Prevalence | 95% CI | $I^2$ | Events | Sample size | Analysis Software | Model | Appraisal Tool | AMSTAR-2 Quality | Remarks |
|---|---|---|---|---|---|---|---|---|---|---|---|---|---|---|
| | | Anxiety | All | 31 | 32.7 | 29.9–38.2 | 98 | 12340 | 37736 | | | | | |
| | | Depression | All | 23 | 25.8 | 20.4–31.0 | 98 | 9347 | 36230 | | | | | |
| | | Sleep Disturbance | All | 8 | 37.3 | 32.1–42.8 | 98 | 3649 | 9784 | | | | | |
| 22 | 69 | Stress | All | 41 | 44.9 | 37–31.1 | 99.8 | 37170 | 82783 | STATA | REM | STROBE | Low | HCPs |
| (Mahmud et al. 2021) [66] | | Anxiety | All | 75 | 41.4 | 36.2–46.7 | 99.8 | 61038 | 147435 | | | | | |
| | | Depression | All | 69 | 37.1 | 31.8–42.4 | 99.8 | 53665 | 144649 | | | | | |
| | | Sleep Disturbance | All | 21 | 43.8 | 35.8–51.7 | 99.4 | 14616 | 33370 | | | | | |
| 23 | 70 | Stress | All | 9 | 31.4 | 17.5–47.3 | 99.8 | 7979 | 25412 | OMA | REM | AHRQ | Moderate | HCPs |
| (Marvaldi et al. 2021) [67] | | Anxiety | All | 22 | 30 | 24.2–37 | 99.6 | 15583 | 51942 | | | | | |
| | | Depression | All | 25 | 31.1 | 25.7–36.8 | 996 | 21157 | 68030 | | | | | |
| | | Sleep Disturbance | All | 10 | 44 | 24.6–64.5 | 99.8 | 5468 | 12428 | | | | | |
| 24 | 83 | Stress | All | 19 | 23.2 | 10.5–35.9 | 99.1 | 10666 | 45976 | STATA | REM | NOS | Moderate | Mixed |
| (Phiri et al. 2021) [70] | | Anxiety | All | 67 | 21.9 | 18.7–25 | 97.8 | 24296 | 110940 | | | | | |
| | | | Nurses | 17 | 27 | 20–34 | 97.8 | 1856 | 6875 | | | | | |
| | | | Physicians | 13 | 17 | 12–22 | 95.6 | 880 | 5177 | | | | | |
| | | Depression | All | 67 | 23.4 | 20.6–26.3 | 99.5 | 28699 | 122644 | | | | | |
| | | Sleep Disturbance | All | 17 | 24 | 16.4–31.6 | 99.5 | 9626 | 40110 | | | | | |
| 25 | 71 | Anxiety | All | 71 | 25 | 21–29 | 99.8 | 14641 | 58565 | STATA | REM | JBI | Low | HCPs |
| (Santabárbara et al. 2021) [73] | | | | | | | | | | | | | | |
| 26 | 38 | Stress | All | 14 | 37 | 25–50 | 99.8 | 5695 | 15391 | STATA | REM | JBI | Low | HCPs |
| (Saragih et al. 2021) [77] | | Anxiety | All | 33 | 40 | 29–52 | 99.8 | 12453 | 31132 | | | | | |
| | | Depression | All | 30 | 37 | 29–45 | 99.7 | 16738 | 45238 | | | | | |
| 27 | 6 | Stress | All | 6 | 65.1 | 48.2–80.3 | 98.6 | 1577 | 2423 | STATA | REM | NIH | Critically Low | Mixed/India |
| (Singh et al. 2021) [76] | | Anxiety | All | 5 | 35.3 | 26.3–44.9 | 92.3 | 658 | 1863 | | | | | |

(Continued)

Table 1. (Continued)

| | Study | Total Study Included | Outcome | Population | Study Included | Prevalence | 95% CI | $I^2$ | Events | Sample size | Analysis Software | Model | Appraisal Tool | AMSTAR-2 Quality | Remarks |
|---|---|---|---|---|---|---|---|---|---|---|---|---|---|---|---|
| | | | Depression | All | 6 | 35.4 | 25.1–46.4 | 97.6 | 1534 | 4333 | | | | | |
| 28 | (Sun et al. 2021) [77] | 47 | Anxiety | All | 44 | 37 | 31–42 | 99.9 | 23828 | 64401 | STATA | REM | NOS | Low | HCPs |
| | | | | Nurses | 9 | 34 | 26–41 | 99 | 9310 | 27382 | | | | | |
| | | | | Physicians | 9 | 28 | 19–38 | 99.1 | 7667 | 27382 | | | | | |
| | | | Depression | All | 39 | 39 | 31–41 | 99.6 | 30395 | 77936 | | | | | |
| | | | | Nurses | 8 | 38 | 30–46 | 99 | 10339 | 27209 | | | | | |
| | | | | Physicians | 8 | 33 | 26–40 | 97.9 | 8979 | 27209 | | | | | |
| | | | Sleep Disturbance | All | 9 | 32 | 23–42 | 99.5 | 4040 | 12626 | | | | | |
| 29 | (Varghese et al. 2021) [79] | 26 | Stress | Nurses | 10 | 40.6 | 25.6–56.8 | 98.6 | 1707 | 4204 | OMA | REM | Loney | Low | Nurses |
| | | | Anxiety | Nurses | 21 | 33 | 24–43 | 99.4 | 4502 | 13641 | | | | | |
| | | | Depression | Nurses | 17 | 32 | 21–44 | 99.4 | 3934 | 12294 | | | | | |
| 30 | (Wu et al. 2021) [95] | 29 | Stress | All | 5 | 41.2 | 19.8–64.5 | 99.8 | 4188 | 10165 | STATA | REM | STROBE | Moderate | Mixed |
| | | | Anxiety | All | 23 | 29 | 23.6–34.7 | 99.4 | 14541 | 50143 | | | | | |
| | | | Depression | All | 23 | 31 | 24.7–37.5 | 99.5 | 12986 | 41889 | | | | | |
| | | | Sleep Disturbance | All | 7 | 47.3 | 38.8–55.8 | 98.7 | 6326 | 13375 | | | | | |
| 31 | (Xia et al. 2021) [80] | 17 | Sleep Disturbance | All | 15 | 45.1 | 37.2–53.1 | 98.7 | 5530 | 12261 | STATA | REM | Loney | Moderate | HCPs/China |
| 32 | (Deng et al. 2021) [55] | 34 | Anxiety | All | 22 | 40 | 33–46 | 99 | 4560 | 11401 | R | REM | AHRQ | Low | HCPs/China |
| | | | Depression | All | 20 | 31 | 25–37 | 98.4 | 3546 | 11438 | | | | | |
| 33 | (Dong et al. 2021) [56] | 22 | Stress | All | 9 | 29.1 | 24.3–33.8 | 96.9 | 4258 | 14631 | STATA | REM | AHRQ | Moderate | HCPs/China |
| | | | Anxiety | All | 22 | 34.4 | 29.5–39.4 | 98.8 | 10338 | 30052 | | | | | |
| | | | Depression | All | 18 | 31.1 | 24.5–37.7 | 99.2 | 6827 | 21953 | | | | | |
| 34 | (Serrano-Ripoll et al. 2021) [75] | 13 | Sleep Disturbance | All | 13 | 38 | 37–39 | 99 | 5349 | 14075 | STATA | REM | MU | Moderate | HCPs |
| 35 | (Hossain et al. 2021) [62] | 17 | Anxiety | All | 15 | 43.6 | 33.1–54.5 | 99.2 | 6557 | 15038 | STATA | REM | NOS | Low | Mixed/South Asia |
| | | | Depression | All | 14 | 29.9 | 23.9–36.6 | 98.1 | 4984 | 16670 | | | | | |
| 36 | (Li et al. 2021) [64] | 65 | Stress | All | 9 | 21.5 | 10.5–34.9 | 99.7 | 5254 | 24439 | STATA | REM | MU | Low | HCPs |

(Continued)

Table 1. (Continued)

| Study | Total Study Included | Outcome | Population | Study Included | Prevalence | 95% CI | $I^2$ | Events | Sample size | Analysis Software | Model | Appraisal Tool | AMSTAR-2 Quality | Remarks |
|---|---|---|---|---|---|---|---|---|---|---|---|---|---|---|
| 37 (El-Qushayri et al. 2021) [58] | 10 | Anxiety | All | 57 | 22.1 | 18.2–26.3 | 99.4 | 16416 | 74280 | | | | | |
| | | Depression | All | 55 | 21.7 | 18.3–25.2 | 99.3 | 18373 | 84666 | | | | | |
| | | Stress | All | 6 | 66.6 | 47.6–81.3 | 98 | 1273 | 1911 | CMA | REM | NIH | Moderate | HCPs/Egypt |
| | | Anxiety | All | 4 | 71.9 | 49.4–86.9 | 98 | 966 | 1344 | | | | | |
| | | Depression | All | 5 | 65.5 | 46.9–80.3 | 98 | 1090 | 1664 | | | | | |
| 38 (Jahrami et al. 2021) [63] | 11 | Sleep Disturbance | All | 11 | 36 | 21.1–54.2 | 99 | 1747 | 4854 | R | REM | NOS | Moderate | Mixed |
| 39 (Olaya et al. 2021) [69] | 57 | Depression | All | 46 | 24 | 20–25 | 99.3 | 12846 | 53527 | STATA | REM | JBI | Low | HCPs |
| | | | Nurses | 14 | 25 | 18–33 | 97.7 | 1471 | 5883 | | | | | |
| | | | Physicians | 10 | 24 | 16–31 | 96.9 | 1024 | 4266 | | | | | |
| 40 (Raoofi et al. 2021) [71] | 46 | Anxiety | All | 46 | 26.1 | 19–34.6 | 99 | 16065 | 61551 | R | REM | NOS | Low | HCPs |
| | | | Nurses | 31 | 24.7 | 17.6–33.5 | 99 | 3074 | 12447 | | | | | |
| | | | Physicians | 17 | 24 | 12.6–40.9 | 99 | 1239 | 5162 | | | | | |
| 41 (Salehi et al. 2021) [72] | 4 | Stress | All | 4 | 11 | 5–16 | 95 | 217 | 1977 | STATA | REM | STROBE | Moderate | Mixed |
| 42 (Abdulla et al. 2021) [47] | 23 | Stress | All | 12 | 58.1 | 44.8–71.3 | 99 | 2445 | 4209 | Rev | REM | 27-itmes | Moderate | HCPs /India |
| | | Anxiety | All | 10 | 42.9 | 30.3–55.5 | 98 | 1312 | 3059 | | | | | |
| | | Depression | All | 5 | 41.9 | 29.2–54.6 | 99 | 2429 | 5796 | | | | | |
| 43 (Crocamo et al. 2021) [54] | 14 | Depression | All | 14 | 23.8 | 16.2–32.2 | 99 | 3373 | 14173 | STATA | REM | NOS | Low | HCPs |
| 44 (Halemani et al. 2021) [59] | 13 | Stress | Nurses | 4 | 37 | 8–66 | 99.6 | 720 | 1946 | STATA | REM | NOS | Low | HCPs |
| | | | Physicians | 3 | 37 | 6–68 | 99 | 343 | 928 | | | | | |
| | | Anxiety | Nurses | 13 | 42 | 33–50 | 97 | 1750 | 4167 | | | | | |
| | | | Physicians | 12 | 34 | 26–42 | 95 | 1127 | 3315 | | | | | |
| | | Depression | Nurses | 13 | 42 | 32–50 | 97 | 1750 | 4167 | | | | | |
| | | | Physicians | 12 | 34 | 23–45 | 97 | 1127 | 3315 | | | | | |
| | | Sleep Disturbance | Nurses | 4 | 44 | 35–53 | 91 | 637 | 1447 | | | | | |
| | | | Physicians | 4 | 35 | 26–43 | 91 | 500 | 1428 | | | | | |

(Continued)

Table 1. (Continued)

| Study | Total Study Included | Outcome | Population | Study Included | Prevalence | 95% CI | I² | Events | Sample size | Analysis Software | Model | Appraisal Tool | AMSTAR-2 Quality | Remarks |
|---|---|---|---|---|---|---|---|---|---|---|---|---|---|---|
| 45 (Hosen et al. 2021) [61] | 4 | Anxiety | All | 4 | 52 | 32–71 | 96 | 883 | 1699 | STATA | REM | JBI | Low | Mixed/Bangladesh |
| | | Depression | All | 4 | 41 | 35–46 | 62.1 | 697 | 1699 | | | | | |
| 46 (Norhayati et al. 2021) [68] | 80 | Stress | All | 20 | 31.7 | 21.3–42.2 | 98 | 4017 | 12673 | Rev | REM | JBI | Low | HCPs/Asia |
| | | Anxiety | All | 68 | 34.8 | 30.1–38.8 | 100 | 43474 | 124925 | | | | | |
| | | | Nurses | 18 | 36.1 | 23.8–48.4 | 100 | 4178 | 11574 | | | | | |
| | | | Physicians | 18 | 30.1 | 20.6–39.6 | 99 | 2191 | 7279 | | | | | |
| | | Depression | All | 60 | 34.6 | 30.9–38.4 | 100 | 45779 | 132308 | | | | | |
| | | | Nurses | 19 | 36.6 | 27.1–46.2 | 99 | 3718 | 10159 | | | | | |
| | | | Physicians | 19 | 28.3 | 18.9–37.8 | 99 | 1937 | 6845 | | | | | |
| | | Sleep Disturbance | All | 12 | 37.9 | 25.4–50.4 | 100 | 5638 | 14877 | | | | | |
| 47 (Zhao et al. 2021) [83] | | Stress | All | 5 | 28 | 9.5–59 | 99 | 1212 | 4327 | CMA | REM | Loney | Low | Mixed |
| | | Anxiety | All | 14 | 23. | 17.1–30.8 | 98 | 2995 | 13020 | | | | | |
| | | Depression | All | 11 | 23.9 | 15–35.9 | 99 | 2849 | 11922 | | | | | |
| 48 (Thakur and Pathak 2021) [78] | 20 | Stress | All | 3 | 55.6 | 36–110 | 99 | 1356 | 2438 | | REM | NOS | Low | HCPs/Frontline |
| | | Anxiety | All | 10 | 27.2 | 18.1–36.3 | 98 | 3229 | 11873 | | | | | |
| | | Depression | All | 9 | 32 | 18–46 | 99 | 3604 | 11262 | | | | | |
| | | Sleep Disturbance | All | 3 | 34.4 | 32.5–36.3 | 99 | 1296 | 3768 | | | | | |
| 49 (Aymerich et al. 2022) [84] | 239 | Stress | All | 57 | 40 | 32–47 | 99.87 | 19217 | 48042 | STATA | REM | NOS | Low | HCPs |
| | | Anxiety | All | 176 | 42 | 35–48 | 99.94 | 86735 | 206513 | | | | | |
| | | Depression | All | 160 | 33 | 28–38 | 99.95 | 69551 | 210762 | | | | | |
| | | Sleep Disturbance | All | 55 | 42 | 36–48 | 99.57 | 15569 | 37068 | | | | | |
| 50 (Hu et al. 2022) [86] | 71 | Anxiety | All | 60 | 26.7 | 19.6–34.6 | 100 | 20558 | 76998 | R | REM | JBI | Low | HCPs/China |
| | | Depression | All | 56 | 28.7 | 19.6–34.6 | 100 | 25655 | 89390 | | | | | |

(Continued)

Table 1. (Continued)

| Study | Total Study Included | Outcome | Population | Study Included | Prevalence | 95% CI | I² | Events | Sample size | Analysis Software | Model | Appraisal Tool | AMSTAR-2 Quality | Remarks |
|---|---|---|---|---|---|---|---|---|---|---|---|---|---|---|
| 51 (Johns et al. 2022) [88] | 55 | Sleep Disturbance | All | 25 | 40.4 | 33.8–47.2 | 99 | 10883 | 26937 | | | | | Physicians |
| | | Anxiety | Physicians | 30 | 25.8 | 20.4–31.5 | 99.2 | 8586 | 33281 | R | REM | JBI | Low | |
| | | Depression | Physicians | 26 | 20.5 | 16–25.3 | 98.9 | 6447 | 31447 | | | | | |
| 52 (Li et al. 2022) [89] | 48 | Sleep Disturbance | All | 48 | 46.4 | 40.3–52.5 | 99.5 | 15322 | 33021 | STATA | REM | AHRQ | Low | Mixed/China |
| 53 (Rezaei et al. 2022) [90] | 24 | Depression | All | 24 | 26 | 18–35 | | 10923 | 42010 | R | REM | NOS | Low | HCPs |
| 54 (Xiong et al. 2022) [93] | 44 | Stress | All | 7 | 27 | 16–38 | 99.8 | 7330 | 27148 | STATA | REM | AHRQ | Low | HCPs/China |
| | | Anxiety | All | 18 | 17 | 13–21 | 99.5 | 5915 | 34793 | | | | | |
| | | Depression | All | 15 | 15 | 13–16 | 94.3 | 7293 | 48621 | | | | | |
| | | Sleep Disturbance | All | 5 | 15 | 7–23 | 99 | 857 | 5711 | | | | | |
| 55 (Zhang et al. 2022) [94] | 88 | Anxiety | All | 88 | 33.6 | 29.7–37.6 | 99.2 | 25222 | 75066 | STATA | REM | STROBE | Low | HCPs |
| 56 (Ślusarska et al. 2022) [91] | 23 | Anxiety | Nurses | 22 | 29 | 18–40 | 99.9 | 12488 | 43062 | R | REM | AHRQ | Low | HCPs |
| | | Depression | Nurses | 18 | 22 | 15–30 | 99.7 | 8675 | 39430 | | | | | |
| 57 (Blasco-Belled et al. 2022) [85] | 74 | Stress | All | 24 | 32 | 23–42 | 100 | 8122 | 25382 | MetaXL | REM | JBI | Moderate | Mixed |
| | | Anxiety | All | 66 | 31 | 27–36 | 99 | 22309 | 71966 | | | | | |
| | | Depression | All | 60 | 31 | 26–35 | 99 | 26269 | 84740 | | | | | |
| 58 (Huang et al. 2022) [87] | 17 | Stress | All | 3 | 17 | 4–34 | 85 | 45 | 265 | CMA | REM | Hoy's | Low | HCPs/Frontline |
| | | Anxiety | All | 16 | 32 | 20–44 | 98 | 2494 | 7795 | | | | | |
| | | Depression | All | 14 | 31 | 21–41 | 97 | 2326 | 7504 | | | | | |
| 59 (Tran et al. 2022) [92] | 7 | Depression | All | 7 | 17.3 | 9.2–27.3 | 100 | 1555 | 8987 | CMA | REM | JBI | Low | HCPs/Vietnam |
| 60 (Tong et al. 2022) [105] | 19 | Stress | All | 10 | 53 | 41.1–64.9 | 78.3 | 3167 | 5976 | R | REM | NOS | Moderate | HCPs |
| | | Anxiety | All | 19 | 43 | 33.8–52.3 | 99 | 7316 | 17013 | | | | | |
| | | Depression | All | 19 | 44.6 | 36.1–53.1 | 99 | 7588 | 17013 | | | | | |
| | | Sleep Disturbance | All | 16 | 42.9 | 33.9–51.9 | 99 | 6612 | 15413 | | | | | |
| 61 (Andhavarapu et al. 2022) [96] | 119 | Stress | All | 119 | 34 | 30–39 | 90 | 39829 | 117143 | CMA | REM | NOS | Moderate | HCPs |
| 62 (Mamun et al. 2022) [103] | 4 | Sleep Disturbance | All | 4 | 51 | 23–79 | 98 | 575 | 1127 | STATA | REM | JBI | Moderate | HCPs/Bangladesh |

(Continued)

Table 1. (Continued)

| Study | Total Study Included | Outcome | Population | Study Included | Prevalence | 95% CI | I² | Events | Sample size | Analysis Software | Model | Appraisal Tool | AMSTAR-2 Quality | Remarks |
|---|---|---|---|---|---|---|---|---|---|---|---|---|---|---|
| 63 (Cheung et al. 2022) [98] | 6 | Anxiety | All | 6 | 37.8 | 28.7–46.9 | 96 | 1230 | 3253 | STATA | REM | JBI | Moderate | HCPs/Asia |
| | | Depression | All | 6 | 39.8 | 29–50.5 | 97 | 1295 | 3253 | | | | | |
| 64 (Hasen et al. 2023b) [100] | 8 | Depression | All | 7 | 40 | 23–57 | 99 | 1101 | 2752 | STATA | REM | NOS | Moderate | HCPs/ Ethiopia |
| | | Sleep Disturbance | All | 3 | 37 | 13–58 | 98 | 334 | 904 | | | | | |
| 65 (Athe et al. 2023) [97] | 11 | Stress | All | 6 | 50.4 | 22.6–78.2 | 99 | 585 | 1161 | RevMan | REM | | Critically Low | HCPs/India |
| | | Anxiety | All | 10 | 42.9 | 30.3–55.5 | 98 | 1302 | 3036 | | | | | |
| | | Depression | All | 8 | 35.4 | 24.5–46.3 | 97 | 1114 | 3147 | | | | | |
| 66 (Sialakis et al. 2023) [107] | 14 | Anxiety | All | 14 | 41.4 | 30.3–52.9 | 99 | 3221 | 7780 | MedCalc | REM | JBI | Moderate | HCPs |
| | | Depression | All | 14 | 33.8 | 24.7–43.6 | 98 | 2630 | 7780 | | | | | |
| 67 (Gheshlagh et al. 2023) [99] | 12 | Anxiety | All | 12 | 23 | 18–27 | 98 | 6891 | 29960 | STATA | REM | NOS | Moderate | HCPs/ Asia |
| | | Depression | All | 11 | 20 | 14–27 | 99 | 5890 | 29448 | | | | | |
| 68 (Wang et al. 2023) [106] | 14 | Stress | Nurses | 14 | 65.4 | 55.9–79.9 | 99 | 13902 | 21257 | STATA | REM | NOS | Moderate | HCPs |
| 69 (Khobragade and Agrawal 2023) [101] | 39 | Stress | All | 23 | 43 | 30–56 | 99 | 3494 | 8125 | R | REM | NOS | Critically Low | HCPs/India |
| | | Sleep Disturbance | All | 16 | 35 | 28–44 | 97 | 1741 | 4974 | | | | | |
| 70 (Lee et al. 2023) [102] | 107 | Stress | All | 107 | 25.5 | 22.5–28.6 | 100 | 47070 | 184588 | R | REM | JBI | Moderate | HCPs |
| | | | Nurses | 42 | 27.4 | 22.5–32.5 | 99 | 10016 | 36554 | | | | | |
| | | | Physicians | 42 | 22.4 | 16.4–29.1 | 99 | 5635 | 25155 | | | | | |
| | | Anxiety | All | 272 | 28.7 | 26.5–31 | 99 | 84831 | 295578 | | | | | |
| | | | Nurses | 85 | 31.5 | 26.9–36.3 | 99 | 22899 | 72695 | | | | | |
| | | | Physicians | 81 | 26.9 | 23–31 | 98 | 10072 | 37443 | | | | | |
| | | Depression | All | 274 | 28.5 | 26.5–30.7 | 99 | 94019 | 329891 | | | | | |
| | | | Nurses | 98 | 28 | 24.5–31.7 | 99 | 24327 | 86881 | | | | | |

(Continued)

**Table 1.** (Continued)

| Study | Total Study Included | Outcome | Population | Study Included | Prevalence | 95% CI | I² | Events | Sample size | Analysis Software | Model | Appraisal Tool | AMSTAR-2 Quality | Remarks |
|---|---|---|---|---|---|---|---|---|---|---|---|---|---|---|
| | | | Physicians | 89 | 25.3 | 21.8–29 | 99 | 10417 | 41175 | | | | | |
| | | Sleep Disturbance | All | 54 | 24.4 | 19.4–29.9 | 99 | 7536 | 30886 | | | | | |
| | | | Nurses | 14 | 26 | 16–37.3 | 99 | 1644 | 6324 | | | | | |
| | | | Physicians | 15 | 16 | 10.2–22.7 | 95 | 632 | 3948 | | | | | |
| 71 (Sharma et al. 2023) [104] | | Stress | All | 22 | 36 | 23.7–48.2 | 99 | 2875 | 7985 | R | OMA | JBI | Moderate | HCPs/India |
| | | Anxiety | All | 20 | 25 | 18.4–31.6 | 98 | 1953 | 7811 | | | | | |
| | | Depression | All | 21 | 20.1 | 15.6–24.6 | 97 | 2055 | 10222 | | | | | |
| | | Sleep Disturbance | All | 6 | 18.9 | 9.9–28 | 95 | 296 | 1565 | | | | | |
| 72 (Hasen et al. 2023a) [108] | 13 | Stress | All | 9 | 51 | 41–62 | 98 | 1929 | 3815 | STATA | REM | NOS | Moderate | HCPs/Ethiopia |
| | | Anxiety | All | 8 | 46 | 30–61 | 99 | 1802 | 3703 | | | | | |

National Institutes of Health: NIH; Joanna Briggs Institute (JBI); OMA: Open Meta Analyst;GRADE: Grading of Recommendations Assessment, Development and Evaluations; NOS: Newcastle–Ottawa Scale; AHRQ: Agency for Healthcare Research and Quality; MMAT: Mixed Methods Appraisal Tool; MU: McMaster University; SAQOR: Systematic Assessment of Quality in Observation Research; CMA: Comprehensive Meta-Analysis software; HCPs: Healthcare Professionals.

| Study | Event | Total | Proportion | 95% C.I. | Weights |
|---|---|---|---|---|---|
| (Salehi et al. 2021) | 217 | 1977 | 10.98 | [9.67; 12.43] | 2.4% |
| (Huang et al. 2022) | 45 | 265 | 16.98 | [12.92; 21.99] | 2.2% |
| (Cénat et al. 2021) | 1444 | 6878 | 20.99 | [20.05; 21.97] | 2.4% |
| (Li et al. 2021) | 5254 | 24439 | 21.50 | [20.99; 22.02] | 2.4% |
| (Phiri et al. 2021) | 10666 | 45976 | 23.20 | [22.82; 23.59] | 2.4% |
| (Allan et al. 2020) | 970 | 4147 | 23.39 | [22.13; 24.70] | 2.4% |
| (Lee et al. 2023) | 47070 | 184588 | 25.50 | [25.30; 25.70] | 2.4% |
| (Hao et al. 2021) | 790 | 3085 | 25.61 | [24.10; 27.18] | 2.4% |
| (Xiong et al. 2022) | 7330 | 27148 | 27.00 | [26.48; 27.53] | 2.4% |
| (Zhao et al. 2021) | 1212 | 4327 | 28.01 | [26.69; 29.37] | 2.4% |
| (Dong et al. 2021) | 4258 | 14631 | 29.10 | [28.37; 29.84] | 2.4% |
| (Salazar de Pablo et al. 2020) | 2030 | 6789 | 29.90 | [28.82; 31.00] | 2.4% |
| (Liu et al. 2021) | 979 | 3200 | 30.59 | [29.02; 32.21] | 2.4% |
| (Marvaldi et al. 2021) | 7979 | 25412 | 31.40 | [30.83; 31.97] | 2.4% |
| (Norhayati et al. 2021) | 4017 | 12673 | 31.70 | [30.89; 32.51] | 2.4% |
| (Blasco-Belled et al. 2022) | 8122 | 25382 | 32.00 | [31.43; 32.58] | 2.4% |
| (Krishnamoorthy et al. 2020) | 1986 | 6017 | 33.01 | [31.83; 34.21] | 2.4% |
| (Andhavarapu et al. 2022) | 39829 | 117143 | 34.00 | [33.73; 34.27] | 2.4% |
| (Sharma et al. 2023) | 2875 | 7985 | 36.01 | [34.96; 37.06] | 2.4% |
| (Ching et al. 2021) | 12380 | 34010 | 36.40 | [35.89; 36.91] | 2.4% |
| (Salari et al. 2020b) | 1293 | 3551 | 36.41 | [34.84; 38.01] | 2.4% |
| (Halemani et al. 2021a) | 343 | 928 | 36.96 | [33.91; 40.12] | 2.4% |
| (Halemani et al. 2021) | 720 | 1946 | 37.00 | [34.88; 39.17] | 2.4% |
| (Saragih et al. 2021) | 5695 | 15391 | 37.00 | [36.24; 37.77] | 2.4% |
| (Dutta et al. 2021) | 10269 | 27238 | 37.70 | [37.13; 38.28] | 2.4% |
| (Aymerich et al. 2022) | 19217 | 48042 | 40.00 | [39.56; 40.44] | 2.4% |
| (Batra et al. 2020) | 6543 | 16235 | 40.30 | [39.55; 41.06] | 2.4% |
| (Varghese et al. 2021) | 1707 | 4204 | 40.60 | [39.13; 42.10] | 2.4% |
| (Wu et al. 2021) | 4188 | 10165 | 41.20 | [40.25; 42.16] | 2.4% |
| (Zhang et al. 2021) | 3866 | 9183 | 42.10 | [41.09; 43.11] | 2.4% |
| (Al Maqbali et al. 2021) | 11625 | 27034 | 43.00 | [42.41; 43.59] | 2.4% |
| (Khobragade and Agrawal 2023) | 3494 | 8125 | 43.00 | [41.93; 44.08] | 2.4% |
| (Mahmud et al. 2021) | 37170 | 82783 | 44.90 | [44.56; 45.24] | 2.4% |
| (Athe et al. 2023) | 585 | 1161 | 50.39 | [47.51; 53.26] | 2.4% |
| (Hasen et al. 2023a) | 1929 | 3815 | 50.56 | [48.98; 52.15] | 2.4% |
| (Tong et al. 2022) | 3167 | 5976 | 53.00 | [51.73; 54.26] | 2.4% |
| (Thakur and Pathak 2021) | 1356 | 2438 | 55.62 | [53.64; 57.58] | 2.4% |
| (Yan et al. 2021) | 428 | 765 | 55.95 | [52.41; 59.43] | 2.4% |
| (Abdulla et al. 2021) | 2445 | 4209 | 58.09 | [56.59; 59.57] | 2.4% |
| (Singh et al. 2021) | 1577 | 2423 | 65.08 | [63.16; 66.96] | 2.4% |
| (Wang et al. 2023) | 13902 | 21257 | 65.40 | [64.76; 66.04] | 2.4% |
| (El-Qushayri et al. 2021) | 1273 | 1911 | 66.61 | [64.47; 68.69] | 2.4% |
| **Random effects model** | | **854852** | **36.95** | **[32.87; 41.22]** | **100.0%** |
| **Prediction interval** | | | | **[14.86; 66.31]** | |

Heterogeneity: $I^2$ = 100%, $\chi^2_{41}$ = 32462.08 ($p$ = 0)

**Fig 2. Forest plot of the prevalence of stress among HCPs stress (N = 42).**

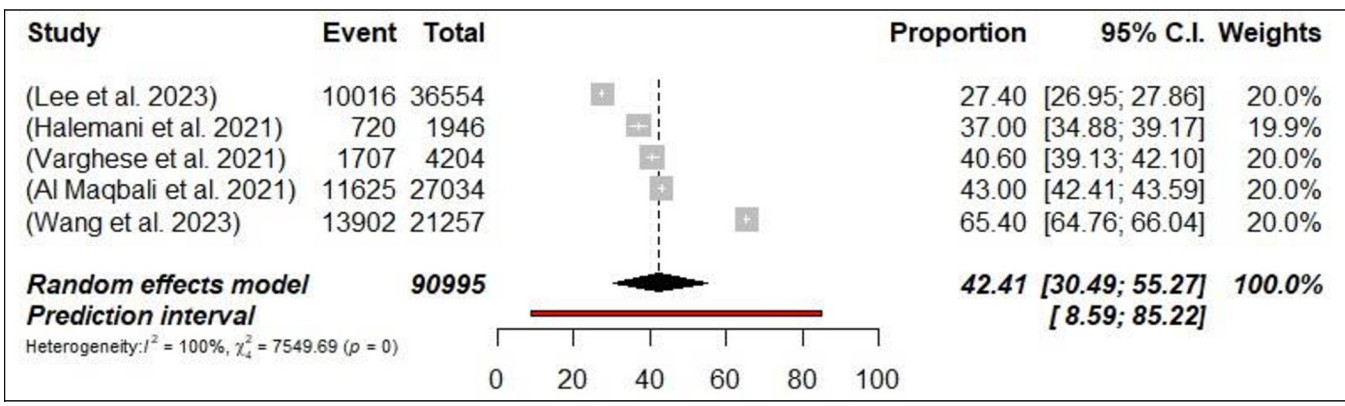

**Fig 3. Forest plot of the prevalence of stress among nurses (N = 5).**

one-out analysis, revealed that none of the meta-analyses had an impact on the global prevalence estimate of anxiety symptoms greater than 1%. Suggestive evidence (class III) was found for the estimated prevalence of anxiety in the case of HCPs, nurses and physicians.

### 3.5 Prevalence of depression

A total of 54 meta-analyses examined the prevalence of depression among HCPs during the COVID-19 pandemic and results ranged from 14% [52] to 65.6% [58]. The pooled prevalence was 29.4% (698,808/2,349,613 participants, 95% CI 27.13–31.84) with 95% PI: 15.01–49.62 (Fig 7: Forest Plots) and there was a significant result in terms of the study heterogeneity ($p <$ 0.0001, $I^2$ = 99.9%). In subgroups analyses, the prevalence of depression was higher among nurses 32% (n = 11; 95% CI = 28–36.35, $I^2$ = 99%) compared with physicians 28.4% (n = 8; 95% CI = 24.32–32.78, $I^2$ = 99%) (Figs 8 and 9: Forest Plots). In the sensitivity analysis, the pooled prevalence remained stable when one meta-analysis was excluded at a time, with

**Table 2. Level of evidence for the prevalence of symptoms among HCPs.**

|  | No | Total | Event | Prevalence % | Random effects (95% CI) | Random effects p-value | 95% Prediction Interval | $I^2$, % | Egger Test | Begg | Class of Evidence | Sensitivity Analysis |
|---|---|---|---|---|---|---|---|---|---|---|---|---|
| **Stress** |  |  |  |  |  |  |  |  |  |  |  |  |
| HCPs | 42 | 854852 | 292245 | 37 | 32.87–41.22 | <0.0001 | 14.86–66.3 | 99.9 | 0.19 | 0.79 | III | ± 2% |
| Nurses | 5 | 90995 | 37970 | 42.4 | 30.49–55.27 | <0.0001 | 8.59–85.22 | 99.9 | 0.87 | 1 | III | ± 3% |
| **Anxiety** |  |  |  |  |  |  |  |  |  |  |  |  |
| HCPs | 55 | 2310774 | 734036 | 31.8 | 29.2–34.61 | <0.0001 | 15.24–54.83 | 99.9 | 0.23 | 0.53 | III | ± 1% |
| Nurses | 12 | 321415 | 105438 | 31.6 | 28.33–35.14 | <0.0001 | 19.54–46.86 | 99.6 | 0.31 | 0.58 | III | ± 1% |
| Physician | 9 | 160937 | 43219 | 26.3 | 22.89–30.10 | <0.0001 | 14.98–42.05 | 99.2 | 0.65 | 1 | III | ± 1% |
| **Depression** |  |  |  |  |  |  |  |  |  |  |  |  |
| HCPs | 54 | 2349613 | 698808 | 29.4 | 27.13–31.84 | <0.0001 | 15.01–49.62 | 99.9 | 0.16 | 0.84 | III | ± 0.5% |
| Nurses | 11 | 305385 | 96564 | 32 | 28–36.35 | <0.0001 | 17.98–50.34 | 99.8 | 0.82 | 0.94 | III | ± 1% |
| Physician | 8 | 154935 | 43268 | 28.4 | 24.32–32.78 | <0.0001 | 15.37–46.32 | 99.8 | 0.98 | 0.62 | III | ± 1% |
| **Sleep** |  |  |  |  |  |  |  |  |  |  |  |  |
| HCPs | 36 | 502780 | 191673 | 36.9 | 33.78–40.05 | <0.0001 | 19.99–57.70 | 99.7 | 0.24 | 0.04 | III | ± 1% |
| Nurses | 5 | 50797 | 19285 | 37.1 | 30.71–44.1 | <0.0001 | 30.71–44.06 | 99.3 | 0.33 | 0.61 | III | ± 2% |
| Physician | 4 | 12976 | 4138 | 30.6 | 20.04–43.77 | <0.0001 | 2.68-87-62 | 99.5 | 0.48 | 1 | III | ± 3% |

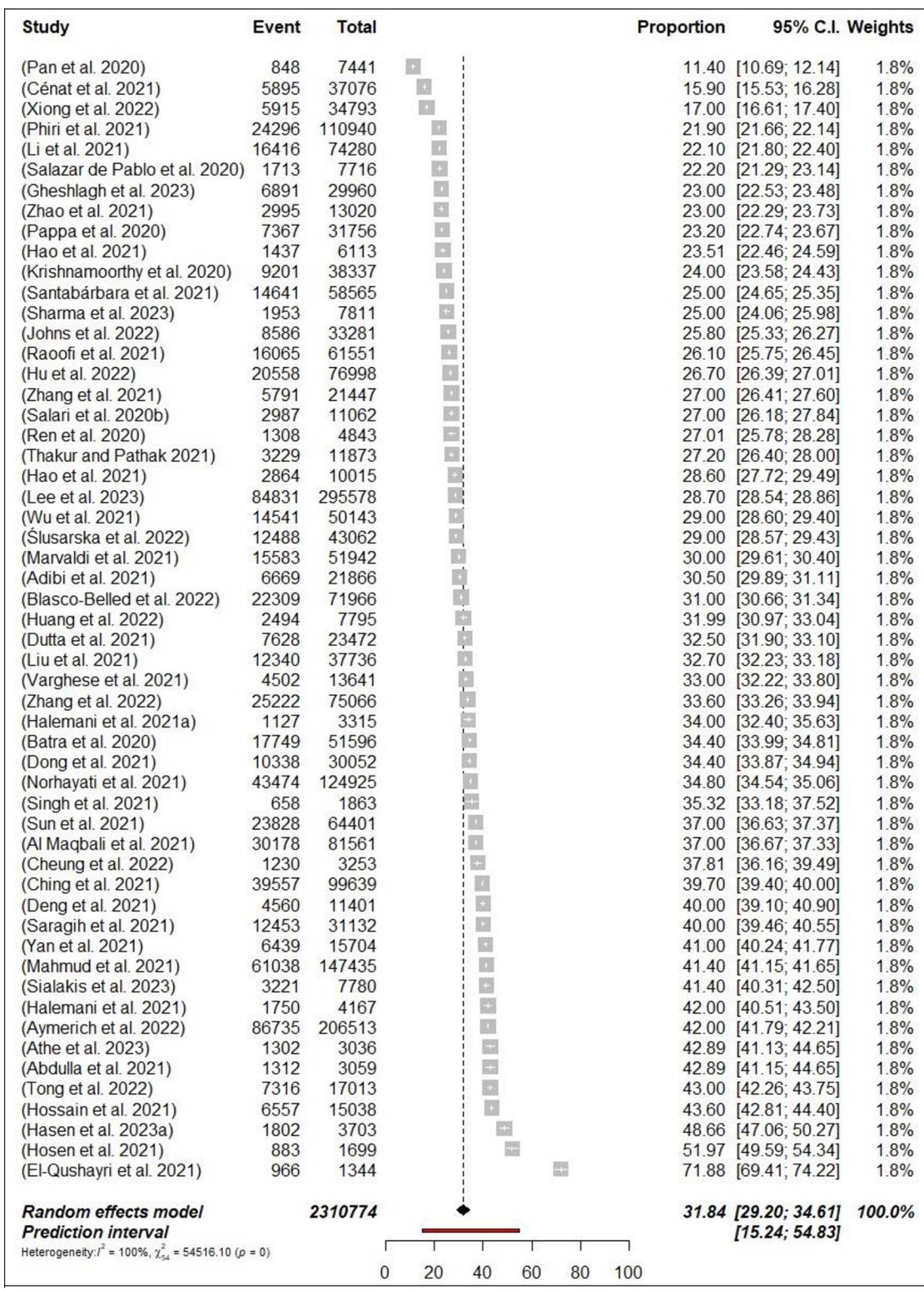

**Fig 4. Forest plot of the prevalence of anxiety among HCPs (N = 55).**

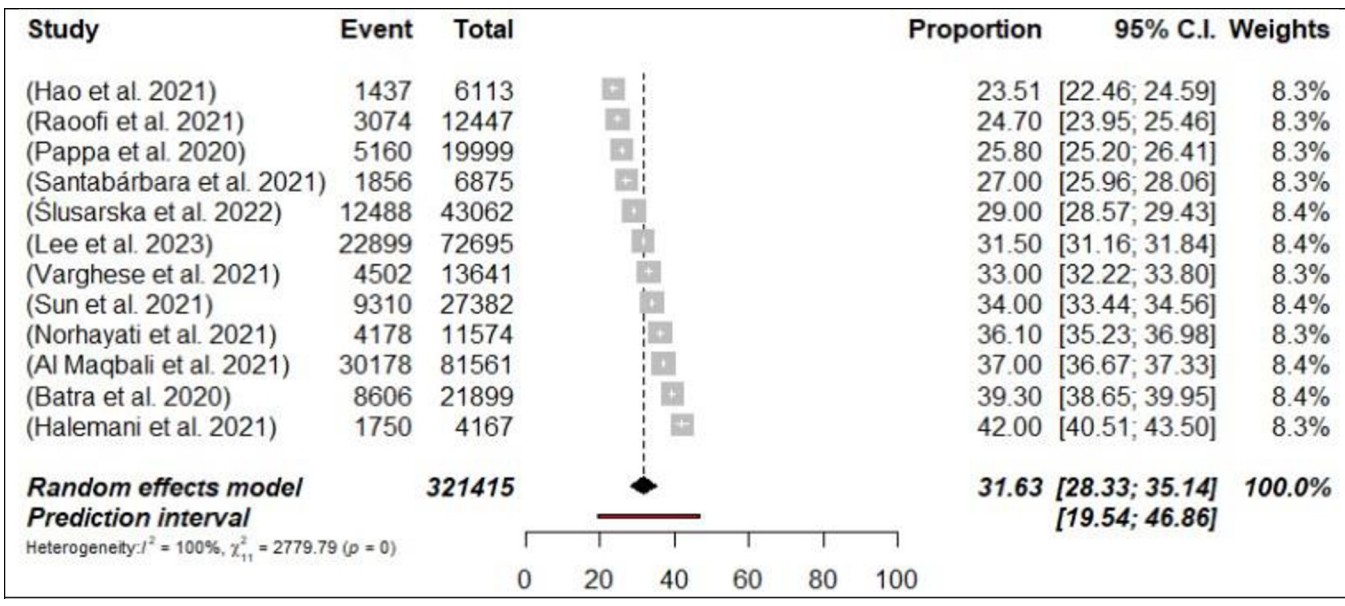

**Fig 5. Forest plot of the prevalence of anxiety among nurses (N = 12).**

variations of less than 1%. Class III evidence revealed suggestive findings regarding the estimated prevalence of depression among HCPs, nurses, and physicians.

### 3.6 Prevalence of sleep disturbance

Sleep disturbance was assessed in 36 meta-analyses, with a calculated pooled prevalence of 36.9% (191,673/502,780 participants, 95% CI 33.78–40.05) with 95% PI: 19.99–57.70 (Fig 10: Forest Plots) with significant differences in terms of the meta-analyses heterogeneity presented (p< 0.0001, $I^2$ = 99.7%). The prevalence of sleep disturbance ranged from 15.01% [93] to 47.3% [95]. In subgroup analyses, the prevalence of sleep disturbance was found to be higher among nurses at 37.1% (n = 5; 95% CI = 30.71–44.1, $I^2$ = 99%) compared to physicians, where

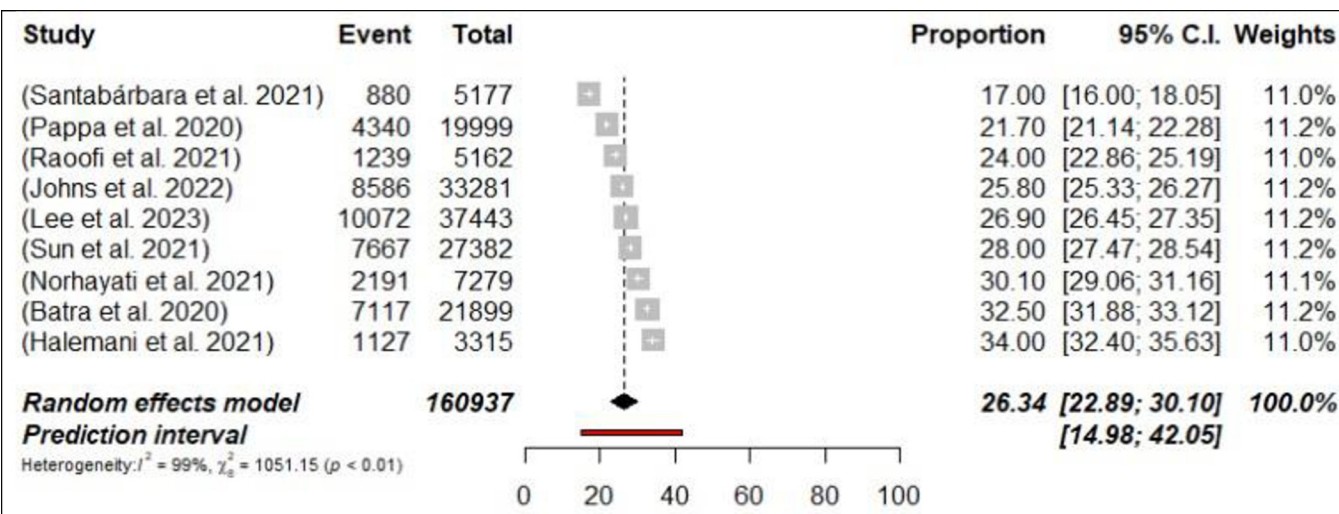

**Fig 6. Forest plot of the prevalence of anxiety among physicians (N = 9).**

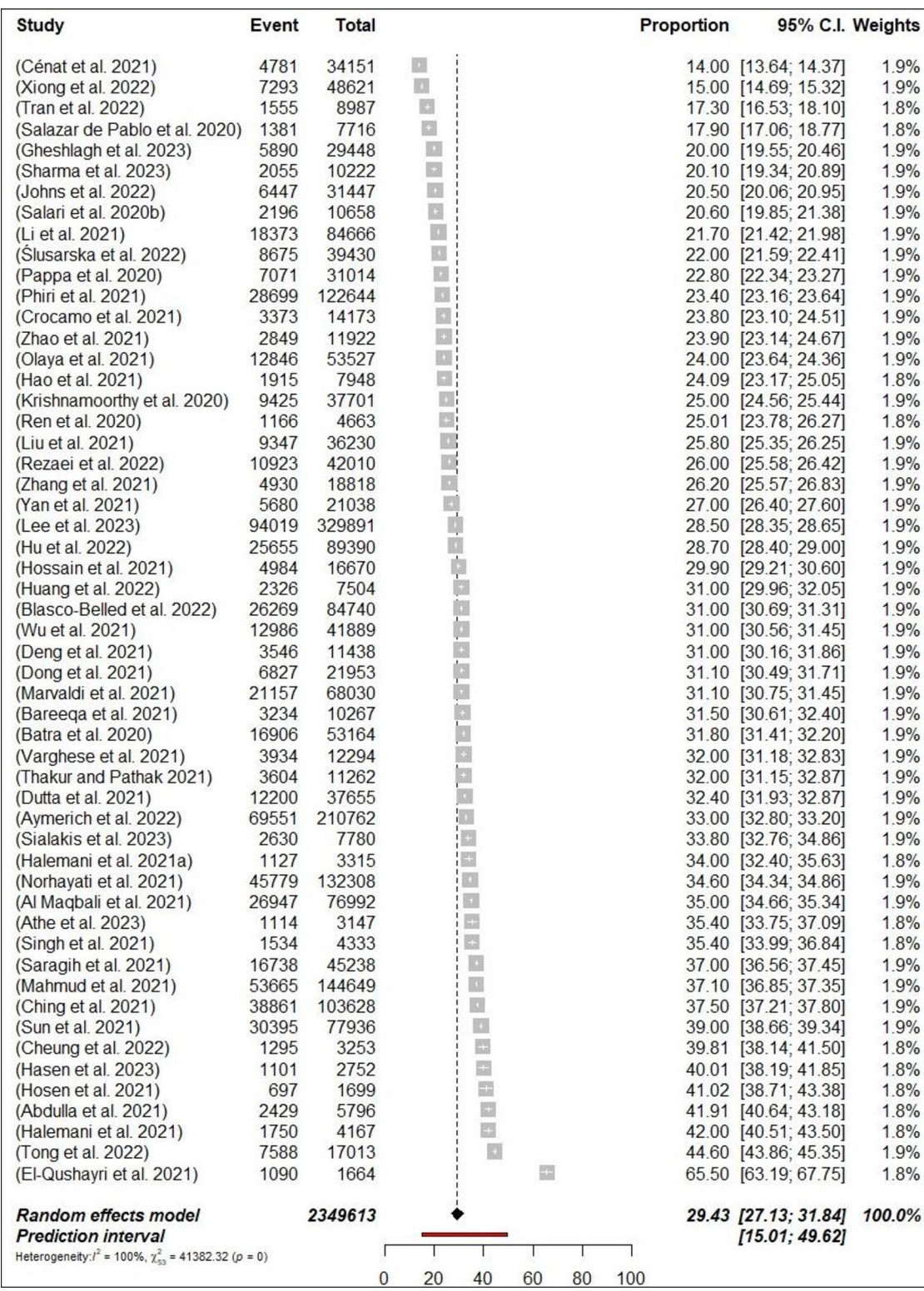

**Fig 7. Forest plot of the prevalence of depression among HCPs (N = 54).**

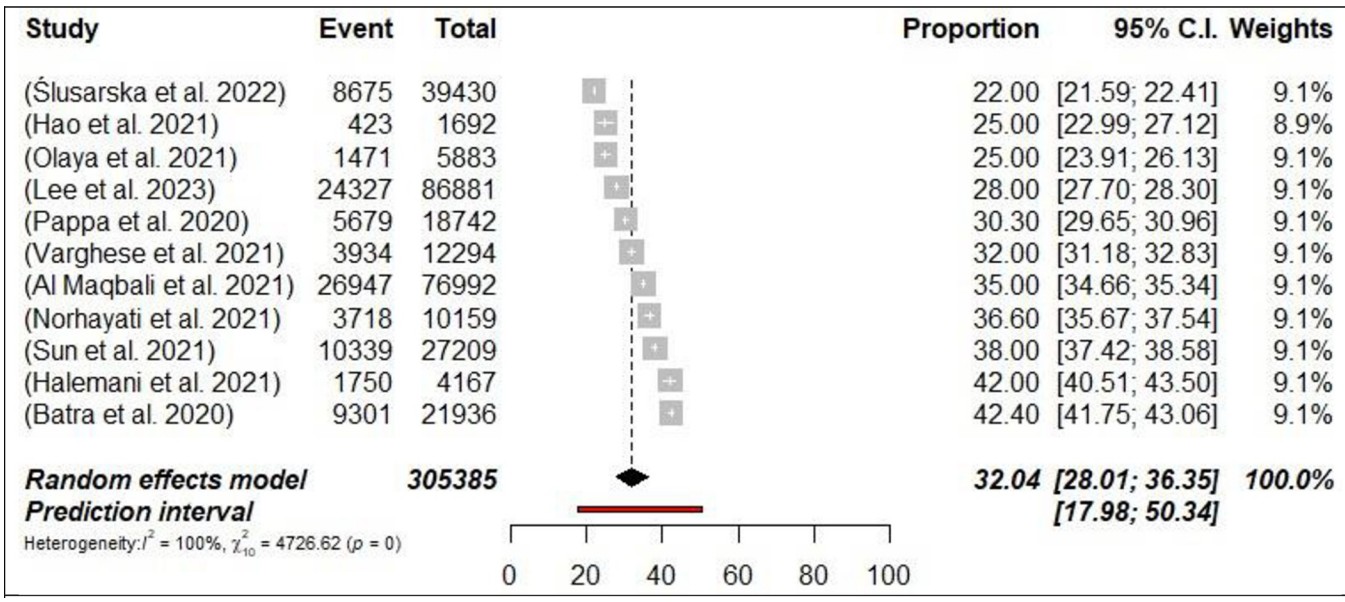

**Fig 8. Forest plot of the prevalence of depression among nurses (N = 11).**

it was 30.6% (n = 4; 95% CI = 20.04–43.77, $I^2$ = 99%) (Figs 11 and 12: Forest Plots). The estimated prevalence rate of sleep disturbance was deemed to be suggestive evidence (Class III). The pooled prevalence did not change in sensitivity analysis by excluding one meta-analyses each time by less than 3%.

## 3.7 Publication bias

The result of Egger's regression test for all pooled prevalence indicates that publication bias was insignificant, showing no evidence of publication bias Table 2.

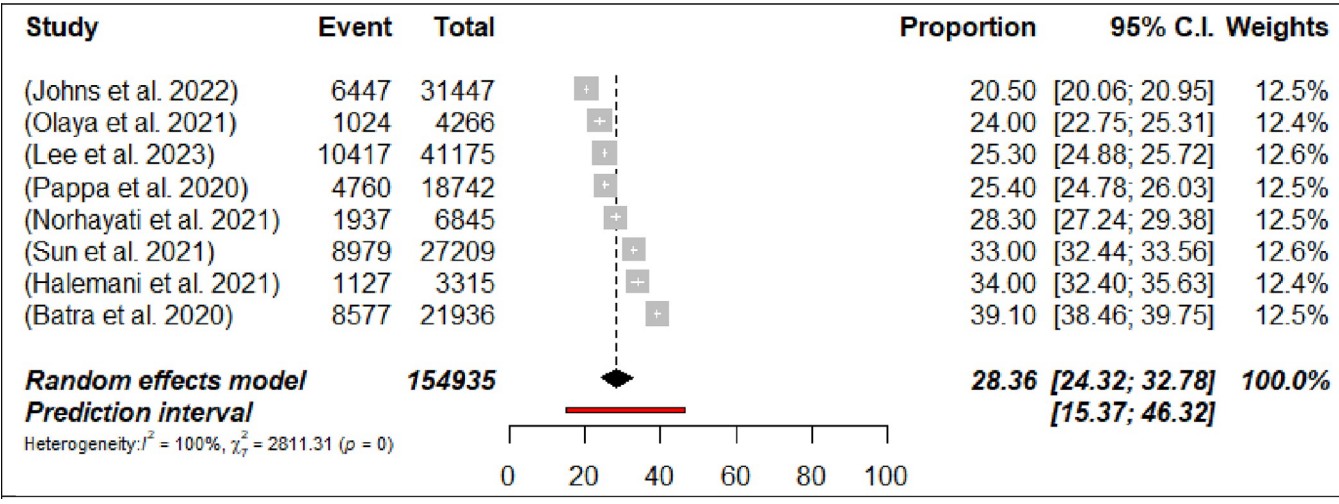

**Fig 9. Forest plot of the prevalence of depression among physicians (N = 8).**

| Study | Event | Total | Proportion | 95% C.I. | Weights |
|---|---|---|---|---|---|
| (Xiong et al. 2022) | 857 | 5711 | 15.01 | [14.10; 15.96] | 2.8% |
| (Cénat et al. 2021) | 993 | 6209 | 15.99 | [15.10; 16.93] | 2.8% |
| (Sharma et al. 2023) | 296 | 1565 | 18.91 | [17.05; 20.93] | 2.7% |
| (Phiri et al. 2021) | 9626 | 40110 | 24.00 | [23.58; 24.42] | 2.8% |
| (Lee et al. 2023) | 7536 | 30886 | 24.40 | [23.92; 24.88] | 2.8% |
| (Sun et al. 2021) | 4040 | 12626 | 32.00 | [31.19; 32.82] | 2.8% |
| (Pappa et al. 2020) | 2935 | 8558 | 34.30 | [33.30; 35.31] | 2.8% |
| (Thakur and Pathak 2021) | 1296 | 3768 | 34.39 | [32.89; 35.93] | 2.8% |
| (Salari et al. 2020a) | 2526 | 7259 | 34.80 | [33.71; 35.90] | 2.8% |
| (Khobragade and Agrawal 2023) | 1741 | 4974 | 35.00 | [33.69; 36.34] | 2.8% |
| (Halemani et al. 2021a) | 500 | 1428 | 35.01 | [32.58; 37.53] | 2.7% |
| (Zhang et al. 2021) | 4174 | 11758 | 35.50 | [34.64; 36.37] | 2.8% |
| (Jahrami et al. 2021) | 1747 | 4854 | 35.99 | [34.65; 37.35] | 2.8% |
| (Dutta et al. 2021) | 3810 | 10411 | 36.60 | [35.68; 37.53] | 2.8% |
| (Hasen et al. 2023) | 334 | 904 | 36.95 | [33.86; 40.14] | 2.7% |
| (Liu et al. 2021) | 3649 | 9784 | 37.30 | [36.34; 38.26] | 2.8% |
| (Norhayati et al. 2021) | 5638 | 14877 | 37.90 | [37.12; 38.68] | 2.8% |
| (Serrano-Ripoll et al. 2021) | 5349 | 14075 | 38.00 | [37.20; 38.81] | 2.8% |
| (Qiu et al. 2020) | 12446 | 31749 | 39.20 | [38.67; 39.74] | 2.8% |
| (Hu et al. 2022) | 10883 | 26937 | 40.40 | [39.82; 40.99] | 2.8% |
| (Yan et al. 2021) | 3065 | 7476 | 41.00 | [39.89; 42.12] | 2.8% |
| (Salari et al. 2020a.) | 2286 | 5496 | 41.59 | [40.30; 42.90] | 2.8% |
| (Aymerich et al. 2022) | 15569 | 37068 | 42.00 | [41.50; 42.50] | 2.8% |
| (Tong et al. 2022) | 6612 | 15413 | 42.90 | [42.12; 43.68] | 2.8% |
| (Krishnamoorthy et al. 2020) | 2010 | 4675 | 42.99 | [41.58; 44.42] | 2.8% |
| (Alimoradi et al. 2021) | 25521 | 59350 | 43.00 | [42.60; 43.40] | 2.8% |
| (Al Maqbali et al. 2021) | 4600 | 10697 | 43.00 | [42.07; 43.94] | 2.8% |
| (Mahmud et al. 2021) | 14616 | 33370 | 43.80 | [43.27; 44.33] | 2.8% |
| (Marvaldi et al. 2021) | 5468 | 12428 | 44.00 | [43.13; 44.87] | 2.8% |
| (Halemani et al. 2021) | 637 | 1447 | 44.02 | [41.48; 46.59] | 2.7% |
| (Hao et al. 2021) | 1607 | 3643 | 44.11 | [42.51; 45.73] | 2.8% |
| (Salazar de Pablo et al. 2020) | 1553 | 3490 | 44.50 | [42.86; 46.15] | 2.8% |
| (Xia et al. 2021) | 5530 | 12261 | 45.10 | [44.22; 45.98] | 2.8% |
| (Li et al. 2022) | 15322 | 33021 | 46.40 | [45.86; 46.94] | 2.8% |
| (Wu et al. 2021) | 6326 | 13375 | 47.30 | [46.45; 48.14] | 2.8% |
| (Mamun et al. 2022) | 575 | 1127 | 51.02 | [48.10; 53.93] | 2.7% |
| **Random effects model** | | **502780** | **36.86** | **[33.78; 40.05]** | **100.0%** |
| **Prediction interval** | | | | **[19.99; 57.70]** | |

Heterogeneity: $I^2 = 100\%$, $\chi^2_{35} = 12288.05$ ($p = 0$)

**Fig 10. Forest plot of the prevalence of sleep disturbance among HCPs (N = 36).**

## 4. Discussion

To the best of our knowledge, this is the first umbrella review to provide a comprehensive synthesis of the estimate of the aggregate data prevalence symptoms of stress, anxiety, depression, and sleep disturbance among HCPs, physicians, and nurses during the entire COVID-19 pandemic.

In the present umbrella review, which utilizes aggregate data from 71 meta-analyses, the most prevalent problems among healthcare professionals (HCPs) were found to be stress

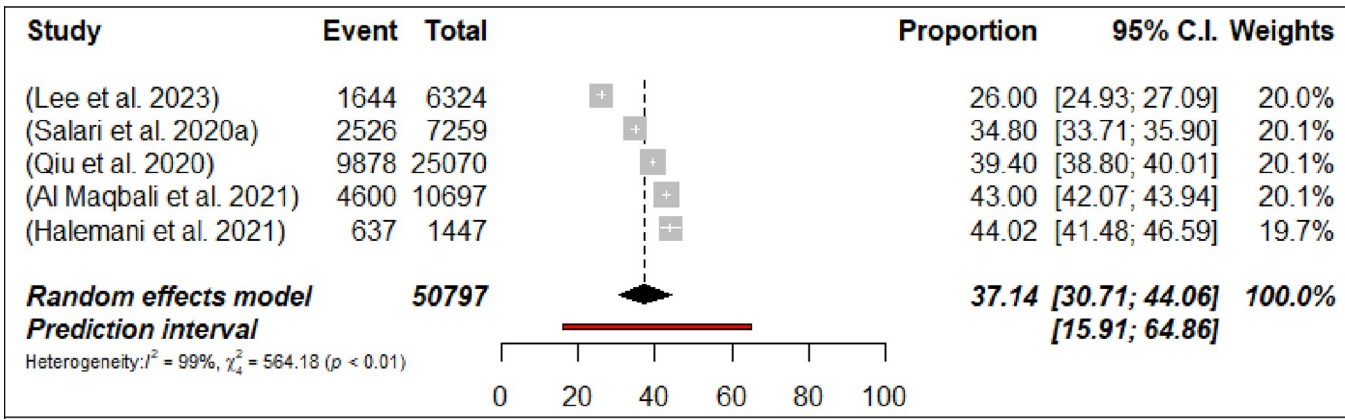

**Fig 11. Forest plot of the prevalence of sleep disturbance among nurses (N = 5).**

(37%), followed by sleep disturbance (36.9%), anxiety (31.8%), and depression (29.4%). The findings among HCPs are slightly higher than the prevalence estimates from the general population (prevalence estimates of 27% for sleep disturbance; 36% for stress; 26% for anxiety and 28% for depression) during the COVID-19 pandemic [109]. In addition, a report by the World Health Organization prior to the COVID-19 pandemic estimated that the global prevalence of anxiety and depression was 4.4% and 3.6% respectively [110]. While some of this disparity may result from different methodological approaches used, the prevalence of depression and anxiety during the COVID-19 pandemic appears to have been higher than before the outbreak. The rise in mental health problems among HCPs may have been triggered by the uncertainty surrounding the pandemic, increased workload, and the fear of family transmission, any, or all of which may also contribute to the higher prevalence of these conditions.

The results of this umbrella review revealed higher prevalence rates compared with two previous reviews of meta-analyses [18,20]. These include 10 meta-analyses which reported prevalence rates during COVID-19 among HCPs: 25% for anxiety and 24% for depression [18]. Another umbrella review involving 18 meta-analyses found stress in 36% of the sample, depression in 26%, anxiety in 27% and sleep disturbance in 32% among HCPs during the COVID-19 pandemic [20]. It is important to highlight that the previous reviews included meta-analyses published before March 2021, while this current review included studies published until January 2024. As a result, the current umbrella review includes more meta-analyses

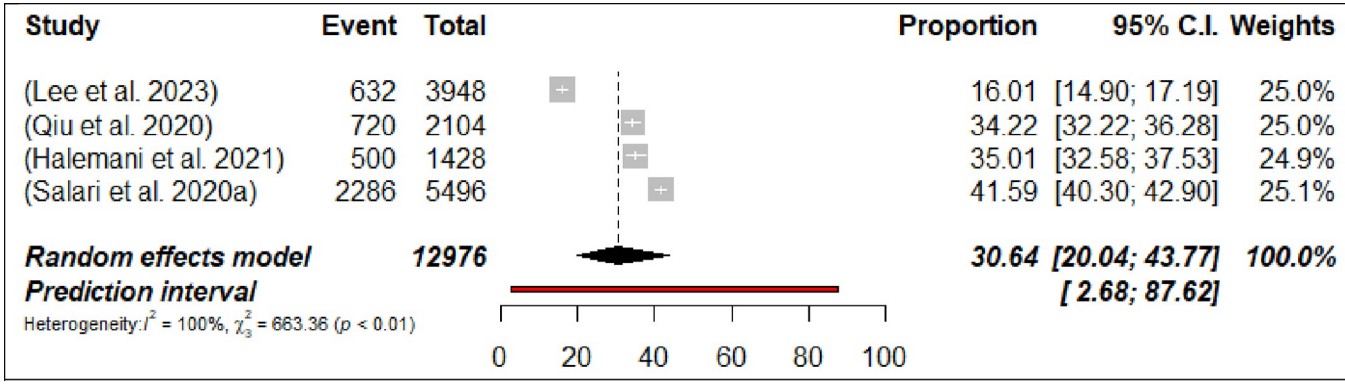

**Fig 12. Forest plot of the prevalence of sleep disturbance among physicians (N = 4).**

compared to the two previous umbrella reviews [18,20]. This umbrella review therefore extends the scientific knowledge of the impact of COVID on mental health of HCPs.

The results of our study suggest that the psychological trauma experienced by HCPs during the SARS and MERS epidemics was lower than that experienced during the COVID-19 pandemic [111–114]. However, the difference between COVID-19 and previous pandemics could be explained by the high mortality rate and infectious potential of the COVID-19 pandemic. The results of this study suggest that the COVID-19 pandemic has had a significant negative effect on the psychological health of HCPs. One important lesson that should be learned is that early detection and treatment are carried out to prevent these types of psychological issues developing into more complex ones.

The result of this analysis demonstrates a higher level of anxiety, depression and sleep disturbance among nurses compared to physicians. One explanation for this may be that nurses are involved in more prolonged and closer contact to COVID-19 patients than physicians [115–117]. Another possible reason might be due to the higher number of nurses included in original studies.

The current review found that the overall pooled prevalence varied between the meta-analyses, for example ranging between 1% [72] and 66.6% [58] for stress, 11.% [40] and 71.9% [58] for anxiety, 14% [52] and 65.6% [58] for depression, and 15% [93] and 47.3% [95] for sleep disturbance. This could be linked to the varying COVID-19 infection and mortality rates in the countries in which the studies were conducted. Other possible reasons might relate to the healthcare system, cultural norms of HCPs, and their perceptions of stress, anxiety, depression, and sleep disturbance which in turn might be influenced by their working conditions, exposure to pandemics, intensity of lockdown and social distancing strategies, and perceived support. For instance, the result of the meta-analysis by El-Qushayri [58] showed the highest prevalent rate in terms of stress, anxiety and depression. This may be because this meta-analysis included only HCPs from Egypt, which in turn might indicate that the Egyptian healthcare system was severely affected by COVID-19 compared to other countries [118].

The finding of this umbrella review highlights significant negative effect that the COVID-19 pandemic has had on the psychological health of HCPs, further emphasizing the need for regular mental health assessment and management in this population. Due to the increasing number of complex traumas that HCPs are experiencing, special attention should be paid to the development of positive traumatic growth. The higher prevalence of stress, anxiety, depression, and sleep disturbance among HCPs have important implications for both the policies and practices of the healthcare system under consideration. It is important to identify effective interventions for HCPs such as the behavioural and educational interventions that have been suggested, including the development of a sense of coherence, positive thinking, and social support [119–121]. Currently there is a lack of evidence about the effectiveness of some psychological interventions that were adapted for use during COVID-19 pandemic specifically for healthcare workers [122,123].

Heterogeneity was significant in the majority of the analyses; several reasons can be attributed to this prominence. Firstly, the individual studies within each meta-analysis might differ substantially in terms of their design, sample sizes, interventions or exposures, and outcome measures. These variations can lead to differing effect sizes or conclusions, making the integration of results into a cohesive summary more complex. Furthermore, heterogeneity can arise from variations in the quality of these studies. While some research might have been meticulously conducted with strict inclusion criteria and rigorous methodologies, other studies on healthcare workers may have inherent biases or confounding factors due to the rapidly changing nature of the pandemic, the pressures of lockdowns, and their effects.

The unique characteristics and experiences of healthcare workers during the COVID-19 crisis, compounded by the challenges of lockdown measures, have the potential to further amplify this variability. Factors such as age, gender, ethnicity, and underlying health conditions, when combined with the stress, increased workload, and challenges of the pandemic and lockdown situations, can significantly influence study outcomes. Additionally, methodological differences in individual studies, like the use of a wide variety of questionnaires to measure symptoms, varied cut-off points, and severity thresholds, as well as the absence of a consistent 'gold standard' for diagnostic interviews, can contribute to increased heterogeneity. In the context of the umbrella review, synthesizing findings from such a diverse collection of meta-analyses, particularly those focused on healthcare workers during this unparalleled period marked by fluctuating lockdown measures, poses a formidable challenge. Such complexity may constrain the robustness and precision of the conclusions drawn.

One of the most critical factors that policymakers need to consider when it comes to implementing effective interventions is the availability of organizational support. This can be done through various work-based interventions such as implementing shorter working hours and having buddy systems [124]. In addition, other measures such as providing mental health consultants and tele counselling can also help reduce the impact of the outbreak of disease on the well-being of staff members [125,126].

### 4.1 Limitations

Several limitations must be taken into consideration when interpreting the results of this umbrella review even though one strength of this methodology is that it provides comprehensive evidence regarding the mental health problems that were faced by HCPs during the COVID-19 pandemic, First, there is a possibility of selection bias. For example, non-English language meta-analyses were not included in this umbrella review, and this may introduce a selection bias. Second, it may be the case that some meta-analyses may have included the same primary studies and that there is consequently a significant study overlap between the meta-analyses included in this review. However, since the results of the studies were then combined with other studies, and a new result was presented, these were regarded as being new studies [25]. Further, several researchers address overlapping by removing some of the reviews with higher rates of overlapping [26,127]. Although removing the overlapping meta-analyses solves the problem of dependent effects, it might introduce a bias of its own. Excluding one of two overlapping meta-analyses from an umbrella review will bias the overall estimate [128,129]. In addition, Hennessy and Johnson [127] clearly mention that the overlap of primary studies included in a meta-review is not necessarily a bias but often can be a benefit.

Third, the various methodologies of the primary studies that were included in the meta-analyses, in terms of sampling methods, assessment tools, operational definitions of the symptoms and study length, might have affected the sensitivity and specificity with regard to detecting the prevalence estimations of stress, anxiety, depression, and sleep disturbance [130]. Finally, it should be noted that stress, anxiety, depression, and sleep disturbance varied between the HCPs studied. Therefore, future research should focus on the difference contexts of estimation prevalence between HCPs and should report the prevalence in each group.

### 5. Conclusion

In summary, this umbrella review systematically analyses the currently available evidence on the prevalence of stress, anxiety, depression, and sleep disturbance among HCPs in relation to COVID-19. It revealed that the incidence of these symptoms is high in the HCP population. However, there is wide variation in the degree of these conditions among this HCP population.

This may be due to the varying experiences of COVID-19 and the cultural differences in the countries where the studies have been carried out. It is clear from the current evidence that strategies involving multi-level interventions are required to develop effective interventions that can help improve the mental health and well-being of HCPs and foster post-traumatic growth. Further research needs to address the limitations of the existing literature, in order to enable the authorities, providers, and patients to improve the quality of mental health on the part of HCPs.

## Supporting information

**S1 Checklist. PRISMA 2020 checklist.**
(DOCX)

**S1 Table. Quality assessment result of meta analysis using the AMSTAR-2 (N = 72).**
(DOCX)

## Author Contributions

**Conceptualization:** Mohammed Al Maqbali, Ahmad Alsayed, Ciara Hughes, Eileen Hacker, Geoffrey L. Dickens.

**Data curation:** Mohammed Al Maqbali, Ahmad Alsayed, Ciara Hughes, Eileen Hacker, Geoffrey L. Dickens.

**Formal analysis:** Mohammed Al Maqbali, Ahmad Alsayed, Ciara Hughes, Eileen Hacker, Geoffrey L. Dickens.

**Funding acquisition:** Mohammed Al Maqbali, Ahmad Alsayed, Ciara Hughes, Eileen Hacker, Geoffrey L. Dickens.

**Investigation:** Mohammed Al Maqbali, Ahmad Alsayed, Ciara Hughes, Eileen Hacker, Geoffrey L. Dickens.

**Methodology:** Mohammed Al Maqbali, Ahmad Alsayed, Ciara Hughes, Eileen Hacker, Geoffrey L. Dickens.

**Project administration:** Mohammed Al Maqbali, Ahmad Alsayed, Ciara Hughes, Eileen Hacker, Geoffrey L. Dickens.

**Resources:** Mohammed Al Maqbali, Geoffrey L. Dickens.

**Software:** Mohammed Al Maqbali, Geoffrey L. Dickens.

**Supervision:** Mohammed Al Maqbali, Geoffrey L. Dickens.

**Validation:** Mohammed Al Maqbali, Geoffrey L. Dickens.

**Visualization:** Mohammed Al Maqbali, Geoffrey L. Dickens.

**Writing – original draft:** Mohammed Al Maqbali, Ahmad Alsayed, Ciara Hughes, Eileen Hacker, Geoffrey L. Dickens.

**Writing – review & editing:** Mohammed Al Maqbali, Ahmad Alsayed, Ciara Hughes, Eileen Hacker, Geoffrey L. Dickens.

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
