## [Decision Letter · Decision Letter 0]

19 Mar 2024

PONE-D-24-02790Stress, Anxiety, Depression and Sleep Disturbance Among Healthcare Professional During the COVID-19 Pandemic: An umbrella review of 71 meta-analysesPLOS ONE

Dear Dr. Dickens,

Thank you for submitting your manuscript to PLOS ONE. After careful consideration, we feel that it has merit but does not fully meet PLOS ONE’s publication criteria as it currently stands. Therefore, we invite you to submit a revised version of the manuscript that addresses the points raised during the review process.

** Be sure to:****reply to the reviewers' comments. **Please submit your revised manuscript by May 03 2024 11:59PM. If you will need more time than this to complete your revisions, please reply to this message or contact the journal office at plosone@plos.org. Please include the following items when submitting your revised manuscript:A rebuttal letter that responds to each point raised by the academic editor and reviewer(s). You should upload this letter as a separate file labeled 'Response to Reviewers'.A marked-up copy of your manuscript that highlights changes made to the original version. You should upload this as a separate file labeled 'Revised Manuscript with Track Changes'.An unmarked version of your revised paper without tracked changes. You should upload this as a separate file labeled 'Manuscript'.If applicable, we recommend that you deposit your laboratory protocols in protocols.io to enhance the reproducibility of your results. Protocols.io assigns your protocol its own identifier (DOI) so that it can be cited independently in the future. For instructions see: https://journals.plos.org/plosone/s/submission-guidelines#loc-laboratory-protocols. Additionally, PLOS ONE offers an option for publishing peer-reviewed Lab Protocol articles, which describe protocols hosted on protocols.io. Read more information on sharing protocols at https://plos.org/protocols?utm_medium=editorial-email&utm_source=authorletters&utm_campaign=protocols.

We look forward to receiving your revised manuscript.

Kind regards,

Fadwa Alhalaiqa

Academic Editor

PLOS ONE

Journal Requirements:

Reviewers' comments:

Reviewer's Responses to Questions

**Comments to the Author**

1. Is the manuscript technically sound, and do the data support the conclusions?

Reviewer #1: Yes

Reviewer #2: Yes

Reviewer #3: Yes

2. Has the statistical analysis been performed appropriately and rigorously? 

Reviewer #1: Yes

Reviewer #2: Yes

Reviewer #3: Yes

3. Have the authors made all data underlying the findings in their manuscript fully available?

Reviewer #1: Yes

Reviewer #2: Yes

Reviewer #3: Yes

4. Is the manuscript presented in an intelligible fashion and written in standard English?

Reviewer #1: Yes

Reviewer #2: Yes

Reviewer #3: Yes

5. Review Comments to the Author

Reviewer #1: This is an interesting paper and the content is very good but I suggest the paper can be improved in the following ways:

-Please correct all parts of the article according to the guidelines of the journal authors guideline

Abstract

-In the conclusion part, it is necessary to specify the researcher's proposal to improve the conditions and use of the beneficiaries

Introduction

Please bring the following items

1- Definition of the research problem

2- The magnitude and importance of the study variable

3- Expressing the necessity of conducting the study

Finally, the practical purpose of the study should be stated

Methods

Please add the qualitative review for selected final manuscript

Discussion

In the discussion section, it is necessary to compare the main results of the study with the results of other studies in this field.

Please refer to published articles and used some reference to citation we proposed this article for enrichment.

-Anxiety, stress and depression levels among nurses of educational hospitals in Iran: Time of performing nursing care for suspected and confirmed COVID-19 patients

-Nursing Students' Competency to Attend Disaster Situations: A Study in Western Iran

- Post-traumatic stress disorder in medical workers involved in earthquake response: A systematic review and meta-analysis

- Prevalence of workplace violence against health care workers in hospital and pre-hospital settings: An umbrella review of meta-analyses

- Providing telenursing care for victims: a simulated study for introducing of possibility nursing interventions in disasters

-What are the strengths and limitations of the study?

Conclusion

What are your suggestion for future studies?

Best regards

Reviewer #2: The authors present an umbrella review to quantify metaanalytic findings aimed at estimating the prevalence of symptoms of stress, anxiety, depression, and sleep disturbance among HCPs during the COVID-19 pandemic. However, clarifications and modifications are needed in the manuscript.

The authors need to clarify whether 71 meta-analyses were included as they stated in the title and at the beginning of the results, or were "Fifty-nine meta-analyses involved 2,308 primary studies were included after a full-text review.", as they stated in the abstract?

Furthermore, in table 1 the authors must add a column where they will list which instruments were used in the 71 systematic reviews included to diagnose the outcomes: stress, anxiety, depression and sleep disorders.

Reviewer #3: Thank you for inviting me to review this interesting umbrella review focused on mental health impacts of COVID-19 pandemic on healthcare professionals. The review is well organized, and it provides comprehensive evidence on the prevalence of mental disorders among HCPs during the COVID-19 pandemic.

A. One of the basic matters in this review is the inclusion of 17 meta-analyses that included a mixed population (General and HCPs-Page 10 line 8) while the remaining studies focus on HCPs only. In this case it is very hard to reach a concrete conclusion for HCPs since significant number of included studies focus on non-healthcare population. Authors need to address this issue at first and most.

B. The number of studies included in the umbrella review mentioned in the title and displayed in the PRISMA diagram (i.e. 71) and stated in the abstract section (i.e. 59) needs correction.

C. A study conducted by Hassen AA and et als (Anxiety and stress among healthcare professionals during COVID-19 in Ethiopia: systematic review and meta-analysis. BMJ Open. 2023, 13 (2) e070367; https://doi.org/10.1136/bmjopen-2022-070367) is missed from your included studies list which could be another finest input for the current review.

D. The study becomes more comprehensive if the factors associated with the mentioned mental health problems were also provided following a meta regression analysis if possible.

Additional revisions needed for further improving the current manuscript are provided below separately for each section.

Abstract

a. Correct spelling for CHINAL,

b. Correct the number (Fifty-nine meta-analyses involved 2,308 primary studies were included after a full-text review.) of meta analyses involved as mentioned in the title and PRISMA diagram.

c. The authors stated that “In subgroup analyses the prevalence of anxiety and depression was higher among nurses than among physicians.” But from the result section the prevalence of sleep disturbance is also higher among nurses than among physicians (Page 12, line 6-8). Therefore, revise the above sentence to “In subgroup analyses the prevalence of anxiety, depression and sleep disturbance was higher among nurses than among physicians.”

Introduction

a. Well described and adequate information included.

Methods

Study selection

a. Authors have mentioned the inclusion and exclusion criteria. BUT what about studies that include both general population and HCPs? From your result section 17 studies included mixed population. (page 10 line 8)

Result

a. Well described

b. Any subgroup analysis for stress??

Slee disturbance

c. Page 12 line 111-12 -“The subgroup analysis was not performed due to an insufficient number of meta-analyses”. Contradicts with the above sentence (page 12 line 6-8) please resolve the paradox.

Discussion and limitations

a. Well, argued.

Annexes

A. Spelling and technical correction needed. (Figure 3. Forest Plot of the Prevalence of Stress Among HCPs Anxiety (N=54)

6. PLOS authors have the option to publish the peer review history of their article (what does this mean?). If published, this will include your full peer review and any attached files.

Reviewer #1: No

Reviewer #2: **Yes: **Ricardo Ney Cobucci

Reviewer #3: No

---

## [Author Response · Author response to Decision Letter 0]

3 Apr 2024

Attached file name (Respond to reviewer)

---

## [Editor Report · Decision Letter 1]

9 Apr 2024

Stress, Anxiety, Depression and Sleep Disturbance Among Healthcare Professional During the COVID-19 Pandemic: An umbrella review of 72 meta-analyses

PONE-D-24-02790R1

Dear Dr. Geoffrey L Dickens,

We’re pleased to inform you that your manuscript has been judged scientifically suitable for publication and will be formally accepted for publication once it meets all outstanding technical requirements.

Kind regards,

Fadwa Alhalaiqa

Academic Editor

PLOS ONE
---

## [Editor Report · Acceptance letter]

26 Apr 2024

PONE-D-24-02790R1 

PLOS ONE

Dear Dr. Dickens, 

I'm pleased to inform you that your manuscript has been deemed suitable for publication in PLOS ONE. Congratulations! Your manuscript is now being handed over to our production team.

Kind regards, 

on behalf of

Pro Fadwa Alhalaiqa 

Academic Editor

PLOS ONE